# ON TRADE-OFFS OF IMAGE PREDICTION IN VISUAL MODEL-BASED REINFORCEMENT LEARNING

## ABSTRACT

Model-based reinforcement learning (MBRL) methods have shown strong sample efficiency and performance across a variety of tasks, including when faced with high-dimensional visual observations. These methods learn to predict the environment dynamics and expected reward from interaction and use this predictive model to plan and perform the task. However, MBRL methods vary in their fundamental design choices, and there is no strong consensus in the literature on how these design decisions affect performance. In this paper, we study a number of design decisions for the predictive model in visual MBRL algorithms, focusing specifically on methods that use a predictive model for planning. We find that a range of design decisions that are often considered crucial, such as the use of latent spaces, have little effect on task performance. A big exception to this finding is that predicting future observations (i.e., images) leads to significant task performance improvement compared to only predicting rewards. We also empirically find that image prediction accuracy, somewhat surprisingly, correlates more strongly with downstream task performance than reward prediction accuracy. We show how this phenomenon is related to exploration and how some of the lower-scoring models on standard benchmarks (that require exploration) will perform the same as the best-performing models when trained on the same training data. Simultaneously, in the absence of exploration, models that fit the data better usually perform better on the downstream task as well, but surprisingly, these are often not the same models that perform the best when learning and exploring from scratch. These findings suggest that performance and exploration place important and potentially contradictory requirements on the model.

## 1 INTRODUCTION

The key component of any model-based reinforcement learning (MBRL) methods is the predictive model. In visual MBRL, this model predicts the future observations (i.e., images) that will result from taking different actions, enabling the agent to select the actions that will lead to the most desirable outcomes. These features enable MBRL agents to perform successfully with high data-efficiency (Deisenroth & Rasmussen, 2011) in many tasks ranging from healthcare (Steyerberg et al., 2019), to robotics (Ebert et al., 2018), and playing board games (Schrittwieser et al., 2019).

More recently, MBRL methods have been extended to settings with high-dimensional observations (i.e., images), where these methods have demonstrated good performance while requiring substantially less data than model-free methods without explicit representation learning (Watter et al., 2015; Finn & Levine, 2017; Zhang et al., 2018; Hafner et al., 2018; Kaiser et al., 2020). However, the models used by these methods, also commonly known as World Models (Ha & Schmidhuber, 2018), vary in their fundamental design. For example, some recent works only predict the expected reward (Oh et al., 2017) or other low-dimensional task-relevant signals (Kahn et al., 2018), while others predict the images as well (Hafner et al., 2019). Along a different axis, some methods model the dynamics of the environment in the latent space (Hafner et al., 2018), while some other approaches model autoregressive dynamics in the observation space (Kaiser et al., 2020).

Unfortunately, there is little comparative analysis of how these design decisions affect performance and efficiency, making it difficult to understand the relative importance of the design decisions that have been put forward in prior work. The goal of this paper is to understand the trade-offs between the design choices of model-based agents. One basic question that we ask is: does predicting images actually provide a benefit for MBRL methods? A tempting alternative to predicting observations is to simply predict future rewards, which, in principle, gives a sufficient signal to infer all task-relevant information. However, as we will see, predicting images has clear and quantifiable benefits – in fact,

we observe that accuracy in predicting observations correlates more strongly with control performance than accuracy of predicting rewards.

Our goal is to specifically analyze the design trade-offs in the *models* themselves, decoupling this as much as possible from the confounding differences in the *algorithm*. While a wide range of different algorithms have been put forward in the literature, we restrict our analysis to arguably the simplest class of MBRL methods, which train a model and then use it for planning *without* any explicit policy. While this limits the scope of our conclusions, it allows us to draw substantially clearer comparisons.

The main contributions of this work are two-fold. First, we provide a coherent conceptual framework for high-level design decisions in creating models. Second, we investigate how each one of these choices and their variations can affect the performance across multiple tasks. We find that:

1. Predicting future observations (i.e. images) leads to significant task performance improvement compared to only predicting rewards. And, somewhat suprisingly, image prediction accuracy correlates more strongly with downstream task performance than reward prediction accuracy.

2. We show how this phenomenon is related to exploration:
   - Some of the lower-scoring models on standard benchmarks that require exploration will perform the same as the best-performing models when trained on the same training data.
   - In the absence of exploration, models that fit the data better usually perform better on the downstream task, but surprisingly, these are often not the same models that perform the best when learning and exploring from scratch.

3. A range of design decisions that are often considered crucial, such as the use of latent spaces, have little effect on task performance.

These findings suggest that performance and exploration place important and potentially contradictory requirements on the model. We will open-source our implementation that can be used to reproduce the experiments and can be further extended to other environments and models.

## 2   RELATED WORK

MBRL is commonly used for applications where sample efficiency is essential, such as real physical systems (Deisenroth & Rasmussen, 2011; Deisenroth et al., 2013; Levine et al., 2016) or healthcare (Raghu et al., 2018; Yu et al., 2019; Steyerberg et al., 2019). In this work, our focus is on settings with a high-dimensional observation space (i.e., images). Scaling MBRL to this setting has proven to be challenging (Zhang et al., 2018), but has shown recent successes (Hafner et al., 2019; Watter et al., 2015; Levine et al., 2016; Finn et al., 2016b; Banijamali et al., 2017; Oh et al., 2017; Zhang et al., 2018; Ebert et al., 2018; Hafner et al., 2018; Dasari et al., 2019; Kaiser et al., 2020).

Besides these methods, a number of works have studied MBRL methods that do not predict pixels, and instead directly predict future rewards (Oh et al., 2017; Liu et al., 2017; Schrittwieser et al., 2019; Sekar et al., 2020), other reward-based quantities (Gelada et al., 2019), or features that correlate with the reward or task (Dosovitskiy & Koltun, 2016; Kahn et al., 2018). It might appear that predicting rewards is sufficient to perform the task, and a reasonable question to ask is whether image prediction accuracy actually correlates with better task performance or not. One of our key findings is that predicting images improves the performance of the agent, suggesting a way to improve the task performance of these methods.

Visual MBRL methods must make a number of architecture choices in structuring the predictive model. Some methods investigate how to make the sequential high-dimensional prediction problem easier by transforming pixels (Finn et al., 2016a; De Brabandere et al., 2016; Liu et al., 2017) or decomposing motion and content (Tulyakov et al., 2017; Denton et al., 2017; Hsieh et al., 2018; Wichers et al., 2018; Amiranashvili et al., 2019). Other methods investigate how to incorporate stochasticity through latent variables (Xue et al., 2016; Babaeizadeh et al., 2018; Denton & Fergus, 2018; Lee et al., 2018; Villegas et al., 2019), autoregressive models (Kalchbrenner et al., 2017; Reed et al., 2017; Weissenborn et al., 2019), flow-based approaches (Kumar et al., 2019) and adversarial methods (Lee et al., 2018). However, whether prediction accuracy actually contributes to MBRL performance has not been verified in detail on image-based tasks. We find a strong correlation between image prediction accuracy and downstream task performance, suggesting video prediction is likely a fruitful area of research for improving visual MBRL.

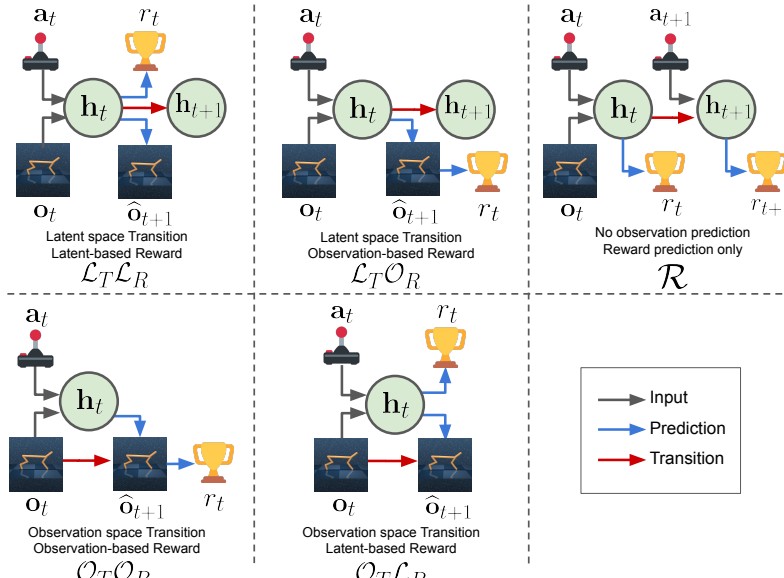

Figure 1: The possible model designs for visual MBRL, based on whether or not to predict images. The rightmost model ($\mathcal{R}$) only predicts the expected rewards conditioned on future actions and previous observations, while other designs predict the images as well. These designs can model the transitions (dynamics) either in the latent space ($\mathcal{L}_{\mathcal{T}}$) or observation space ($\mathcal{O}_{\mathcal{T}}$). The input source for the reward prediction model can be the predicted images ($\mathcal{O}_{\mathcal{R}}$) or the latent state of the model ($\mathcal{L}_{\mathcal{R}}$). Note how these models differ in their training losses (in the case of $\mathcal{R}$) and how the errors are back-propagated, either directly through the latent space or via predicted targets.

## 3    A FRAMEWORK FOR VISUAL MODEL-BASED RL

MBRL methods must model the dynamics of the environment conditioned on the future actions of the agent, as well as current and preceding observations. The data for this model comes from trials in the environment. In an online setting, collecting more data and modeling the environment happens iteratively, and in an offline setting, data collection happens once using a predefined policy. We are interested in cases where the observation state is pixels, which is usually not enough to reveal the exact state of the environment. Since our goal is specifically to analyze the design trade-offs in the *models* themselves, we focus our analysis only on MBRL methods which train a model and use it for planning, rather than MBRL methods that learn policies or value functions. We defer discussion and analysis of policies and value functions to future work.

Therefore, we consider the problem as a partially observable Markov decision process (POMDP) with a discrete time step $t \in [1, T]$, hidden state $\mathbf{s}_t$, observed image $\mathbf{o}_t$, continuous action $\mathbf{a}_t$ and scalar immediate reward $r_t$. The dynamics are defined as a transition function $\mathbf{s}_t \sim p(\mathbf{s}_t \mid \mathbf{s}_{t-1}, \mathbf{a}_{t-1})$, observation function $\mathbf{o}_t \sim p(\mathbf{o}_t \mid \mathbf{s}_t)$ and reward function $r_t \sim p(r_t \mid \mathbf{s}_t)$. In RL, the goal is to find a policy $p(\mathbf{a}_t|\mathbf{o}_{\leq t}, r_{\leq t}, a_{\leq t})$ that maximizes the expected discounted sum of future rewards $\mathbb{E}_p\left[\sum_{t=1}^T \gamma r_t\right]$, where $\gamma$ is the discount factor, $T$ is the task horizon and the expectation is over actions sampled from the policy. In MBRL, we approximate the expected reward by predicting its distribution conditioned on the previous observations and the future actions $p(r_t|\mathbf{o}_{\leq t}, \mathbf{a}_{\geq t})$, and then search for a high-reward action sequences via a policy optimization method such as proximal policy optimization (PPO) (Schulman et al., 2017) or a planning algorithm like the cross-entropy method (CEM) (Rubinstein, 1997; Chua et al., 2018). In this paper we focus on the latter.

Given the assumption that the environment is partially observable and the ground-truth states are not accessible, the models in visual MBRL can be divided into five main categories, depending on training signals and learned representations (Figure 1). The first category approximates the expected rewards conditioned on future actions and previous observations $p(r_t \mid \mathbf{o}_{\leq t}, \mathbf{a}_{\geq t})$ without explicitly predicting images (Oh et al., 2017; Dosovitskiy & Koltun, 2016; Racanière et al., 2017; Liu et al., 2017; Kahn et al., 2018; Schrittwieser et al., 2019). In contrast, the next four categories learn to predict the next observations $\widehat{\mathbf{o}}_{t+1}$, in addition to the reward, which results in a learned latent representation.

Given this latent space, it is possible to: **(1)** Model the transition function of the environment in the observation space: $\widehat{\mathbf{o}}_{t+1} \sim p(\mathbf{o}_t \mid \mathbf{o}_{\leq t}, \mathbf{a}_{\geq t})$ or directly in the latent space: $\mathbf{h}_t \sim p(\mathbf{h}_t \mid \mathbf{h}_{t-1}, \mathbf{a}_t)$ where $\mathbf{h}_t$ is the hidden state of the model at time step $t$ **(2)** Predict the future reward using the learned latent space: $r_t \sim p(r_t \mid \mathbf{h}_t)$ or from the predicted future observation: $r_t \sim p(r_t \mid \widehat{\mathbf{o}}_{t+1})$.

## 4 EXPERIMENTS AND ANALYSIS

Our goal in this paper is to study how each axis of variation in the generalized MBRL framework, described in Section 3, impacts the performance of the agent. To this end, we will not only study end-to-end performance of each model on benchmark tasks, but also analyze how these models compare when there is no exploration (i.e. trained from the same static dataset), evaluate the importance of predicting images, and analyze a variety of other model design choices.

### 4.1 EXPERIMENT SETUP

**Planning Method**   Studying the effects of each design decision independently is difficult due to the complex interactions between the components of MBRL (Sutton & Barto, 2018). Each training run collects different data, entangling exploration and modeling. The added complexity of policy learning further complicates analysis. To isolate the design decisions pertaining to only the predictive model, we focus on MBRL methods that only learn a model, and then plan through that model to select actions (Chua et al., 2018; Zhang et al., 2018; Hafner et al., 2018). We use the cross-entropy method (CEM) (Rubinstein, 1997) to optimize over the actions under a given model, following prior work. We also include an Oracle model in our comparisons, which uses the true simulator to predict the future rewards. Since we are using the same planning algorithm for all the models, the Oracle approximates the maximum possible performance with our planner. Note, however, that given the limited planning horizon of CEM, the Oracle model is not necessarily optimal, particularly in tasks with sparse and far reaching rewards.

**Environments**   We use seven image-based continuous control tasks from the DeepMind Control Suite (Tassa et al., 2018). These environments provide qualitatively different challenges. `cheetah_run` and `walker_walk` exhibit larger state and action spaces, including collisions with the ground that are hard to predict. `finger_spin` includes similar contact dynamics between the finger and the object. `cartpole_balance` require long-term memory because the cart can move out of the frame while `cartpole_swingup` requires long-term planning. `reacher_easy` and `ball_in_cup_catch` also have sparse reward signal, makes them hard to plan for given a short planning horizon. In all tasks, the only observations are $64 \times 64 \times 3$ third-person camera images (visualized in Figure 5 in appendix).

**Implementation and Hyper-Parameter tuning**   Given the empirical nature of this work, our observations are only as good as our implementations. To minimize the effect of hyper-parameters tuning, we chose not to design any new models, and instead used existing high performance well-known implementations. The only exception to this was $\mathcal{R}$, for which we tested multiple architectures and used the best one based on asymptotic performance (see appendix for details). While there are going to be numerous shortcomings for **ANY** implementation, in close collaboration with the original authors of these implementations, we took great care in making these choices so as to minimize the performance effects of implementation details. Moreover, we conducted comprehensive hyper-parameter optimization for `cheetah_run` and relied on the common practice of using the same set of hyper-parameters across other tasks (Hafner et al., 2018). Given all this, we are confident that the performance of our implementations cannot be improved *significantly* on these tasks and for these same methods by further hyper-parameter tuning. Details of these implementations, the planning method, model variations, and hyper-parameters can be found in the appendix.

### 4.2 ONLINE PERFORMANCE

First, we compare the overall asymptotic performance, the stability of training, and the sample efficiency of various model designs across all of the tasks in the online setting. In this setting, after every ten training trajectories, collected with the latest version of the model, we evaluate the agent on one episode. For training episodes, we add random planning noise for better exploration, while in the evaluation episodes there is no additional noise. We train each model for a total of 1000 episodes where each episode is 1000 steps long with an action repeat of four. Therefore, each model *observes* 250K training samples of the environment during its entire training (1M environment steps), while evaluated on a total number of 25K test samples across 1000 evaluation episodes. The possible reward range per environment step is $[0, 1]$; therefore, the maximum possible score on each task is 1000.

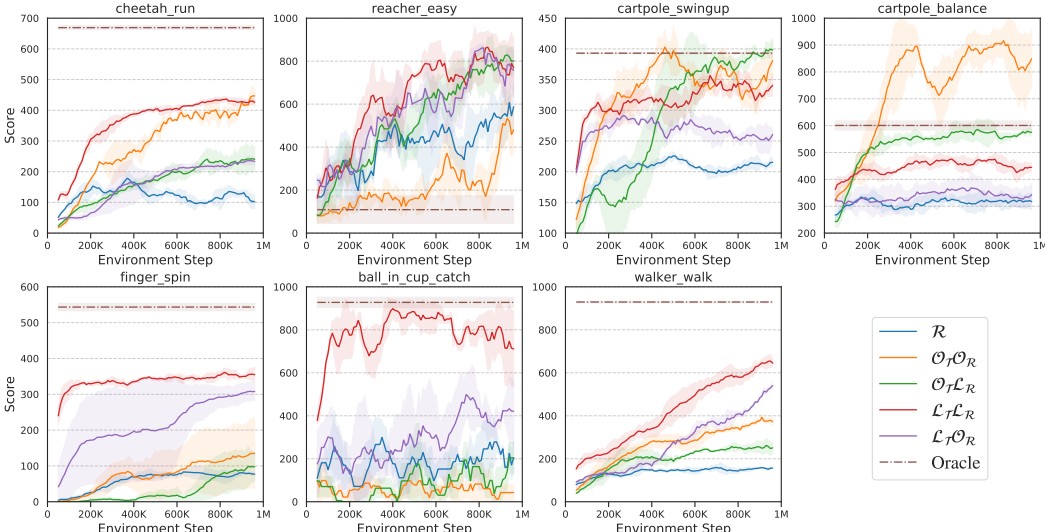

Figure 2: The performance of different models in the online setting. The $x$-axis is the number of environment steps and the $y$-axis is the achieved score by the agent in an evaluation episode. Each curve represents the average score across three independent runs while the shadow is the standard deviation. The model designs are from Figure 1. From this graph, it can be clearly seen that the models which predict images can perform substantially better than the model which only predicts the expected reward ($\mathcal{R}$). In some cases, these models perform even better than the Oracle which suggests that there is not a clear relation between prediction accuracy and performance, given the fact that the Oracle is the perfect predictor. The curves are smoothed by averaging in a moving window of size ten.

The results are shown in Figure 2. There are a number of patterns that we note in these results. First, $\mathcal{R}$ consistently under-performs compared to the rest of the models, attaining lower scores and substantially worse reward distributions in Figure 3. Second, some models clearly perform better than others on each given task e.g. $\mathcal{L}_\mathcal{T}\mathcal{L}_\mathcal{R}$ on ball_in_cup_catch. From this, we might suppose that these models are better able to fit the data, and therefore lead to better rollouts. However, as we will see in the next section, this may not be the case. Third, in some rare cases, some models outperform the Oracle model e.g. $\mathcal{O}_\mathcal{T}\mathcal{O}_\mathcal{R}$ on cartpole_balance. Note, that given the limited planning horizon of CEM, the Oracle model is not necessarily optimal, particularly in tasks with sparse and far reaching rewards such as reacher_easy. This also suggests that there is not a clear relation between prediction accuracy and performance, given the fact that the Oracle is the perfect predictor.

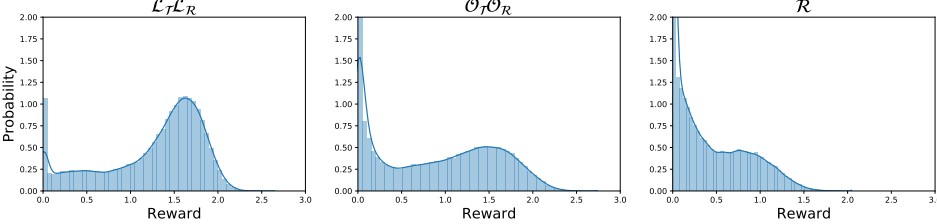

Figure 3: This figure illustrates the distribution of the rewards in trajectories collected by best performing models and $\mathcal{R}$ in the online setting for cheetah_run. Since the rewards for each frame is $[0, 1]$ and the action repeat is set to four, the observed reward is always in $[0, 4]$ ($x$-axis). This graph demonstrates how $\mathcal{L}_\mathcal{T}\mathcal{L}_\mathcal{R}$ and $\mathcal{O}_\mathcal{T}\mathcal{O}_\mathcal{R}$ managed to explore a different subset of state space with higher rewards (compared to $\mathcal{R}$) which directly affected their performance (compare at Figure 2). This shows how entangled exploration and model accuracy are which makes it hard to analyse the effect of each design decision in the online settings.

| | cheetah run | reacher easy | cartpole swingup | cartpole balance | finger spin | ball_in_cup catch | walker walk |
|---|---|---|---|---|---|---|---|
| $\mathcal{R}$ | 216.8 | 996.0 | 258.3 | 415.2 | 171.1 | 985.1 | 299.6 |
| $\mathcal{O_T O_R}$ | **502.6** | 981.1 | **684.7** | **978.5** | 148.0 | 910.3 | 567.5 |
| $\mathcal{O_T L_R}$ | 489.9 | **999.0** | 406.0 | 923.5 | 244.1 | 949.1 | **730.6** |
| $\mathcal{L_T L_R}$ | 502.2 | 994.1 | 413.4 | 543.5 | **373.1** | **996.0** | 552.0 |
| $\mathcal{L_T O_R}$ | 281.7 | 993.0 | 281.0 | 411.0 | 259.3 | 932.0 | 383.3 |

Table 1: The asymptotic performance of various model designs in the offline setting. In this setting there is no exploration and the training dataset is pre-collected and fixed for all the models. The reported numbers are the 90th percentile out of 100 evaluation episodes, averaged across three different runs. This table indicates a significant performance improvement by predicting images almost across all the tasks. Moreover, there is a meaningful difference between the numbers in this table and Figure 2 signifying the importance of exploration in the online settings. Please note how some of best-performing models in this table performed poorly in Figure 2.

| | cheetah run | reacher easy | cartpole swingup | cartpole balance | finger spin | ball_in_cup catch | walker walk |
|---|---|---|---|---|---|---|---|
| $\mathcal{R}$ | 0.2274 | 0.0226 | 0.4152 | 0.1690 | 0.0049 | 0.0055 | 0.2258 |
| $\mathcal{O_T O_R}$ | 0.0432 | 0.0612 | 0.2447 | 0.2175 | 0.0068 | 0.0010 | 0.1779 |
| $\mathcal{O_T L_R}$ | 0.0472 | 0.0280 | 0.2169 | 0.2310 | 0.0099 | 0.0009 | 0.1814 |
| $\mathcal{L_T L_R}$ | 0.0781 | 0.0045 | 0.1183 | 0.1409 | 0.0025 | 0.0019 | 0.1367 |
| $\mathcal{L_T O_R}$ | 0.2022 | 0.0601 | 0.4096 | 0.1649 | 0.0033 | 0.0018 | 0.1984 |
| $\rho(\mathcal{L}_R, \mathcal{S})$ | -0.98 | -0.62 | -0.53 | 0.86 | -0.47 | 0.58 | -0.61 |

Table 2: Median reward prediction error ($\mathcal{L}_R$) of each model across all of the trajectories in evaluation partition of the offline dataset. This table demonstrates a generally better task performance for more accurate models in the offline setting, when compared with Table 1. The last row reports the Pearson correlation coefficient between the reward prediction error and the asymptotic performance for each task across models. This row demonstrates the strong correlation between reward prediction error ($\mathcal{L}_R$) and task performance ($\mathcal{S}$) in the absence of exploration. In cases which all models are close to the maximum possible score of 1000 (such as ball_in_cup_catch) the correlation can be misleading because a better prediction does not help the model anymore.

It is important to emphasize that, in the online setting, agents may explore entirely different regions of the state space, which means they will be trained on different data, as visualized in Figure 3. Because the models are all trained on different data, these results mix the consequences of *prediction accuracy* with the effects of *exploration*. We hypothesize that these two capabilities are distinct, and isolate modeling accuracy from exploration in the following section by training all of the models on the *same* data, and then measuring both their accuracy and task performance. To further understand this complicated relationship between model accuracy, exploration, and task performance, we next compare the different model types when they are all trained on the same data.

### 4.3 Offline Performance

In the offline setting, all of the training trajectories are pre-collected, and therefore all of the models will be trained on the same data. We create this controlled dataset by aggregating all of the trajectories from all of the model variations trained in the online setting (Section 4.2). This dataset contains 1.25M training and 125K testing samples per task. We train each model on this dataset, and then evaluate its performance on the actual task. Table 1 contains the results of this experiment while Figure 8 illustrates the prediction error of each model for cheetah_run.

These results show a very different trend from that seen in Section 4.2. For example, although $\mathcal{O_T L_R}$ was a poor performer in the online setting on walker_walk, it performs substantially better on the same task when all the models are trained from the same data. As another observation, almost all of the models managed to achieve very high scores in the offline setting on reacher_easy and ball_in_cup_catch, while many struggled in the online setting. This suggests that models that result in good performance on a given dataset are *not* necessarily the models that lead to best performance during exploration. We speculate that some degree of model error can result in optimistic

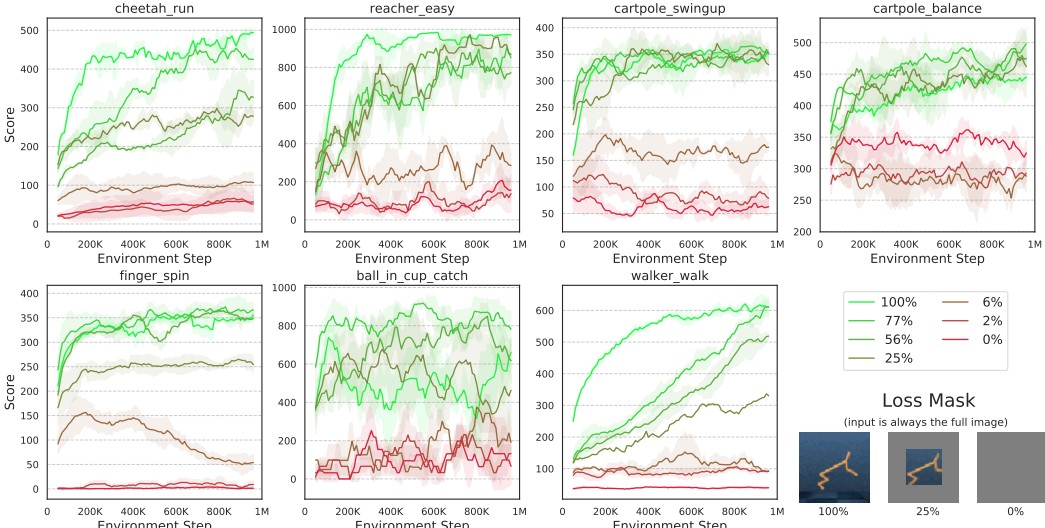

Figure 4: The effect of limiting the image prediction capacity on the performance of the agent. The graphs follow the same format as Figure 2. Each model is a variant of $\mathcal{L}_{\mathcal{T}}\mathcal{L}_{\mathcal{R}}$ which predicts the entire image, or part of it (center cropped) or no image at all. This graph shows that predicting more pixels results in higher performance as well as more stable training and better sample efficiency. Note that in this experiment, the mode still observes the entire image, however, it predicts different number of pixels, forced by a loss mask. The labels indicated what percentage of the image was being predicted by the model.

exploration strategies that make it easier to obtain better data for training *subsequent* models, even if such models do not perform as well on a given dataset as their lower-error counterparts. Please check the appendix for more on this speculation. However, there is a strong correlation between prediction errors in Table 2 and the scores in Table 1, suggesting that in absence of exploration and given the same training data, any model that fits the data better performs the given task better as well.

We would clarify however that our analysis does not necessarily imply that models with worse prediction accuracy result in better exploration, though that is one possible interpretation of the results. Instead, we are observing that simply modeling the current data as accurately as possible does not by itself guarantee good exploration, and models that are worse at prediction might still explore better during planning. We speculate that this can be explained in terms of (implicit) optimism in the face of uncertainty (Auer, 2002) or stochasticity (Osband & Van Roy, 2017; Levine et al., 2020).

### 4.4 THE IMPORTANCE OF IMAGE PREDICTION

The results in Section 4.2 and Section 4.3 differ to a great extent, but there is a common pattern between the two: $\mathcal{R}$ achieves lower scores compared to other variations in both offline and online settings. To provide further empirical support for this observation, we explore the effect of predicting fewer pixels on $\mathcal{L}_{\mathcal{T}}\mathcal{L}_{\mathcal{R}}$. In this experiment, we limit how many pixels $\mathcal{L}_{\mathcal{T}}\mathcal{L}_{\mathcal{R}}$ predicts by center-cropping (Figure 4) and resizing (Figure 6 in appendix) the target image. It is important to note that the model still *observes* the entire image; however, it *predicts* different number of pixels.

These results suggest that prediction of images and its reconstruction actually plays an important role in visual MBRL, and simply substituting reward prediction is not sufficient. Methods that predict images, rather than just reward, perform better in both the online (Section 4.2) and offline (Section 4.3) settings and, as shown in this section, the performance of these models scales with the amount of pixel supervision that they receive. This question raises an immediate question: does the accuracy of image prediction actually corresponds to better task performance? This question is particularly important because predicting the images is computationally expensive, given the fact that observation space is typically high-dimensional.

To explore this correlation, we scale down $\mathcal{O}_{\mathcal{T}}\mathcal{O}_{\mathcal{R}}$ and $\mathcal{L}_{\mathcal{T}}\mathcal{L}_{\mathcal{R}}$ at multiple levels to limit their modeling capacity, and thereby increase their prediction error (details in appendix). Then, we calculate the Pearson correlation coefficient $\rho$ between reward prediction error and observation prediction error to the asymptotic performance of the agent in the online setting. The results of these experiments are summarized in Table 3. This table shows strong correlation between image prediction error and

| Task | $\mathcal{O}_\mathcal{T}\mathcal{O}_\mathcal{R}$ | | | | $\mathcal{L}_\mathcal{T}\mathcal{L}_\mathcal{R}$ | | | |
|---|---|---|---|---|---|---|---|---|
| | $\rho(\mathcal{L}_O,\mathcal{S})$ | $\rho(\mathcal{L}_R,\mathcal{S})$ | $\mu_\mathcal{S}$ | $\sigma_\mathcal{S}$ | $\rho(\mathcal{L}_O,\mathcal{S})$ | $\rho(\mathcal{L}_R,\mathcal{S})$ | $\mu_\mathcal{S}$ | $\sigma_\mathcal{S}$ |
| cheetah_run | -0.96 | -0.09 | 338 | 111 | -0.92 | 0.17 | 566 | 60 |
| cartpole_swingup | -0.94 | -0.69 | 572 | 112 | -0.74 | 0.72 | 503 | 51 |
| cartpole_balance | -0.93 | -0.85 | 877 | 108 | -0.81 | 0.93 | 647 | 93 |
| walker_walk | -0.72 | 0.94 | 308 | 89 | -0.26 | -0.84 | 489 | 149 |
| reacher_easy | -0.36 | 0.63 | 644 | 277 | -0.88 | 0.25 | 992 | 10 |
| finger_spin | -0.08 | 0.93 | 185 | 87 | -0.03 | 0.35 | 417 | 25 |
| ball_in_cup_catch | 0.08 | 0.33 | 925 | 32 | -0.03 | 0.75 | 997 | 3 |

Table 3: Pearson correlation coefficient $\rho$ between image prediction error $\mathcal{L}_O$, reward prediction error $\mathcal{L}_R$ and asymptotic score $\mathcal{S}$. To calculate the correlation, we scaled down $\mathcal{O}_\mathcal{T}\mathcal{O}_\mathcal{R}$ and $\mathcal{L}_\mathcal{T}\mathcal{L}_\mathcal{R}$ at multiple levels to limit their modeling capacity and thereby potentially increase their prediction error. $\mu_\mathcal{S}$ and $\sigma_\mathcal{S}$ are the average and the standard deviation of asymptotic performances across different scales of each model. In cases with low standard deviation of the scores (such as ball_in_cup_catch), meaning all version of the models did more or less the same, the correlation can be misleading. This table demonstrates the strong correlation between image prediction error and task performance.

asymptotic performance. This evidence suggests that MBRL methods can substantially perform better using a better next frame predictive model, especially for more visually complex tasks.

Surprisingly, Table 3 also suggests that reward prediction error does not have a strong correlation to asymptotic performance. Even in some cases, there is a positive correlation between the reward prediction error and the asymptotic performance — meaning the agent performs better when the prediction error is high — which is counter-intuitive. One potential explanation is the same relation between model accuracy and exploration in Section 4.2 and Section 4.3. In some sense, one should *expect* that the reward accuracy correlates negatively with performance, since overfitting on the observed states, combined with pessimism, can harm exploration by guiding the planning algorithm towards familiar states that have high predicted reward, rather than visiting new states that may have higher reward. More related experiments can be found in the appendix.

## 5 CONCLUSIONS

In this paper, we take the first steps towards analyzing the importance of model quality in visual MBRL and how various design decisions affect the overall performance of the agent. We provide empirical evidence that predicting images can substantially improve task performance over only predicting the expected reward. Moreover, we demonstrate how the accuracy of image prediction strongly correlates with the final task performance of these models. We also find models that result in higher rewards from a static dataset may not perform as well when learning and exploring from scratch, and some of the best-performing models on standard benchmarks, which require exploration, do not perform as well as lower-scoring models when both are trained on the same data. These findings suggest that performance and exploration place important and potentially contradictory requirements on the model.

Our results with offline datasets present a complex picture of the interplay between exploration and prediction accuracy. On the one hand, they suggest that simply building the most powerful and accurate models may not necessarily be the right choice for attaining good exploration when learning from scratch. On the other hand, this result can be turned around to imply that models that perform best on benchmark tasks which require exploration may not in fact be the most accurate models. This has considerable implication on MBRL methods that learn from previously collected offline data (Fujimoto et al., 2019; Wu et al., 2019; Agarwal et al., 2020) – a setting that is particularly common for real-world applications e.g. in Robotics (Finn & Levine, 2017).

We would like to emphasize that our findings are limited to the scope of our experiments and our observations may vary under different experimental setup. First, we only trained the models with a fixed number of samples. It would be interesting to investigate whether the benefits of predicting images gets diminished or amplified in lower and higher sample regimes, affecting sample efficiency of these models. Second, using a policy optimization method is a natural next step for expanding these experiments. Finally, it is interesting to investigate whether the findings of this paper can be generalized to more (or less) visually complex tasks in other domains.

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

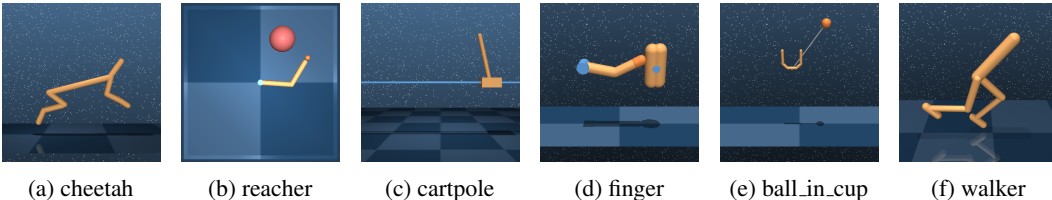

| (a) cheetah | (b) reacher | (c) cartpole | (d) finger | (e) ball_in_cup | (f) walker |

Figure 5: The environments from DeepMind Control Suite (Tassa et al., 2018) used in our experiments. The images show agent observations before downscaling. (a) The `cheetah_run` includes large state and action space. (b) The `reacher_easy` has only a sparse reward. (c) The `cartpole_balance` has a fixed camera so the cart can move out of sight while `cartpole_swingup` requires long term planning. (d) The `finger_spin` includes contacts between the finger and the object. (e) The `ball_in_cup_catch` has a sparse reward that is only given once the ball is caught in the cup. (f) The `walker_walk` has difficult to predict interactions with the ground when the robot is lying down.

## A    ADDITIONAL EXPERIMENTS

### A.1    EFFECT OF RESIZING THE TARGET IMAGE

We repeat the same experiment in Section 4.4, this time by resizing the target instead of cropping it. The results of this experiment, visualized in Figure 6, supports the same observation as Figure 4: the performance of models scales with the amount of pixel supervision that they receive. It is important to note that the model still *observes* the entire image; however, it *predicts* different number of pixels. However, we observe a sharper transition with more pixels when compared with Figure 4. This is expected because resizing the observation preserves more information vs cropping.

### A.2    ON EFFECT OF OPTIMISM

We speculated that some degree of model error can result in optimistic exploration strategies that make it easier to obtain better data for training subsequent models, even if such models do not perform as well on a given dataset as their lower-error counterparts. To expand on this, we compare the models on another carefully designed dataset, which includes only low-reward states in its training set but has trajectories with high reward for evaluation. The rationale is that models that perform well on this evaluation are likely to generalize well to higher-reward states, and thus be more suitable for control. Figure 9 shows a heat-map of ground-truth reward vs. predicted reward for each model on `cheetah_run`. The best performing models on this task in Table 1, $\mathcal{O}_{\mathcal{T}}\mathcal{O}_{\mathcal{R}}$ and $\mathcal{L}_{\mathcal{T}}\mathcal{L}_{\mathcal{R}}$, do not actually exhibit the best generalization. Indeed, they are both pessimistic (very few points in the upper-left half of the plot in the bottom row). In contrast, $\mathcal{L}_{\mathcal{T}}\mathcal{O}_{\mathcal{R}}$ appears to extrapolate well, but performs poorly in Table 1, likely due to the excessively optimistic prediction. At the same time, $\mathcal{R}$ has high error and performed poorly at the same time. One possible implication of this conclusion is that future work should more carefully disentangle and study the different roles of exploration vs. prediction accuracy.

### A.3    EFFECT OF SHARING THE LEARNT LATENT SPACE

In this experiment, we investigate the effect of sharing the learned representation when a model is predicting both reward and images (Figure 7). It is a common belief (Kaiser et al., 2020) that learning a shared representation, typically done by back-propagating from both losses into the same hidden layers, can improve the performance of the agent. However, our results indicate that doing so does not necessarily improve the results, and, particularly in the case of models with observation space dynamics, can result in a large performance drop. This is an interesting observation since most of the prior work, including recent state-of-the-art models such as Hafner et al. (2018) and Kaiser et al. (2020), simply back-propagate all of the losses into a shared learned representation, assuming it will improve the performance.

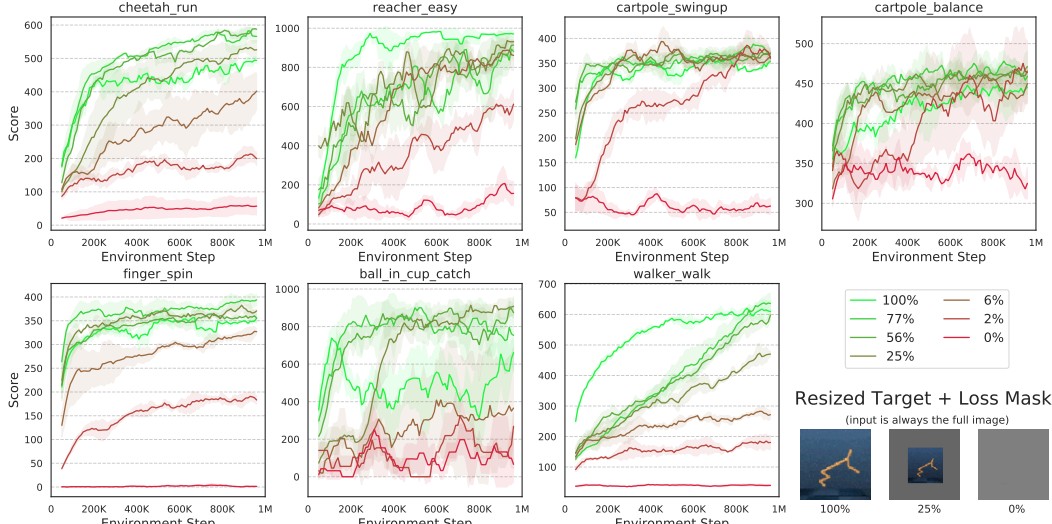

Figure 6: The effect of limiting the observation prediction capacity on the performance of the agent. The setup is exactly the same as Figure 4, except we resized the target image instead of cropping. Similarly, this graph shows that predicting more pixels results in higher performance as well as more stable training and better sample efficiency. Note that in this experiment, all the agents still observe the entire image, however, they predict different number of pixels.

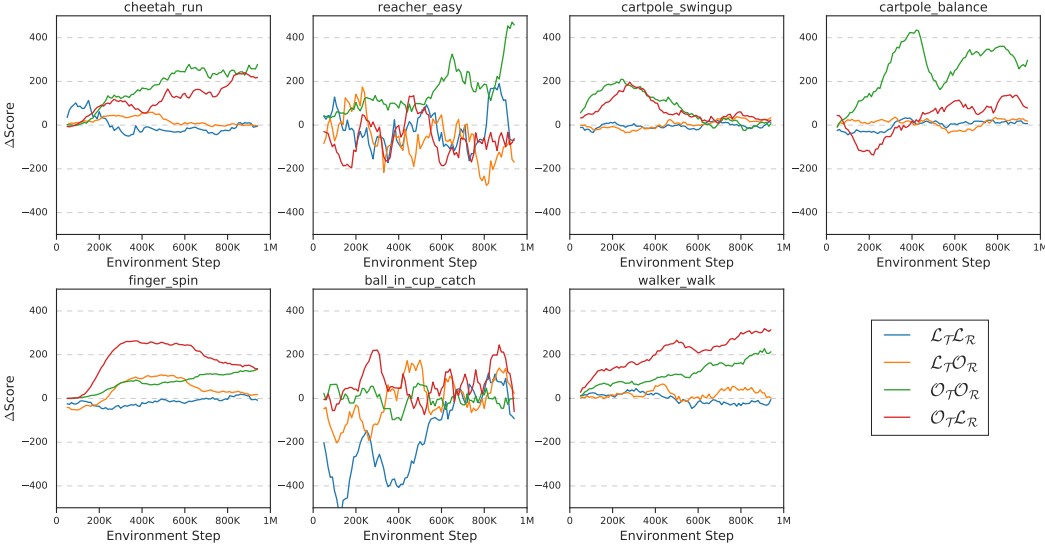

Figure 7: Effect of sharing the learned latent space. The graphs follow the same format as Figure 2 except the $y$-axis is the difference between achieved scores by each model with and without sharing the learned latent space. In other words, a positive delta score on $y$-axis means that **not** sharing the learned latent space is improving the results for each particular model. This figure demonstrates how sharing the learned latent space can degrade the performance, particularly for models with observation space dynamics.

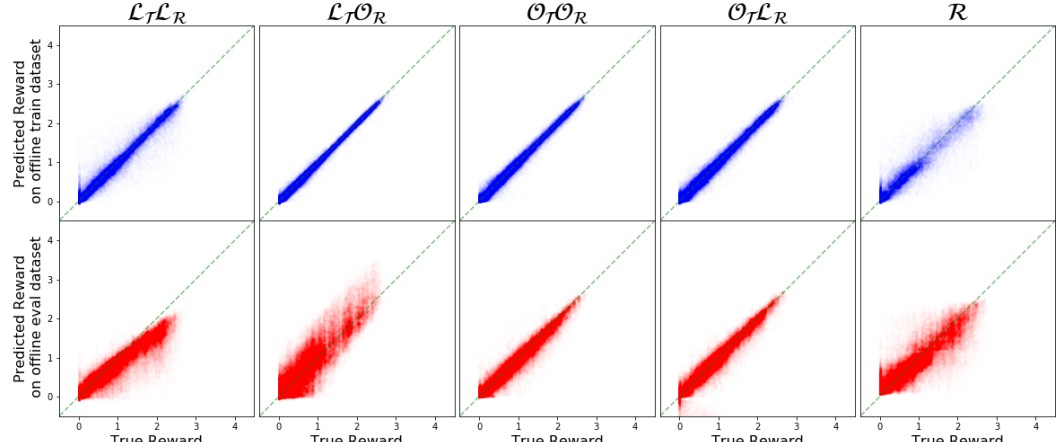

Figure 8: Comparison of training and evaluation accuracy of models with and without image prediction. Each graph demonstrates the heat map of the predicted vs. ground-truth reward for `cheetah_run`. The training and evaluation dataset is fixed for all of the models with fix train/evaluation partitions. Since the rewards for each frame is $[0, 1]$ and the action repeat is set to four, the observed reward is always in $[0, 4]$. The green dash line is the perfect prediction. This figure demonstrates better task performance for more accurate models when compared with Table 1.

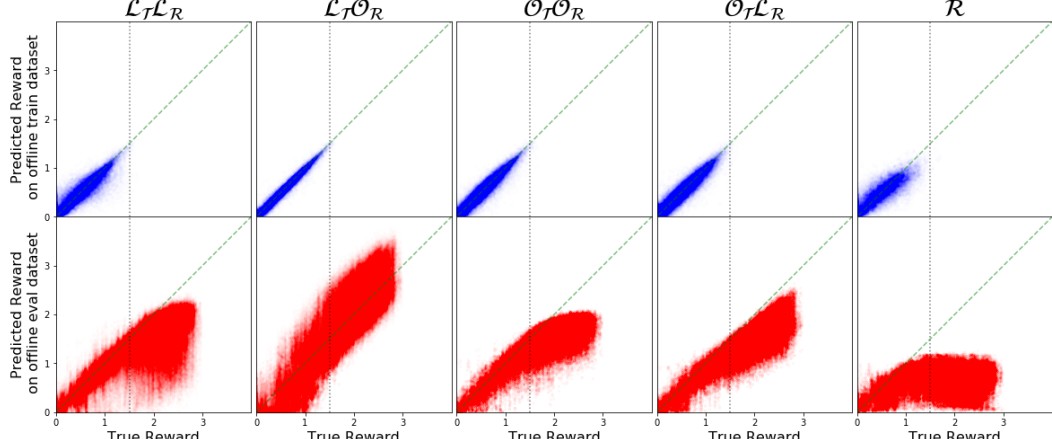

Figure 9: Comparison of training and evaluation accuracy of models with and without image prediction. Each graph demonstrates the heat map of the predicted vs. ground-truth reward for `cheetah_run`. The training and evaluation dataset is fixed for all of the models and designed specifically to make sure that there are unseen high reward states in the evaluation (separated by the dotted black line). Since the rewards for each frame is $[0, 1]$ and the action repeat is set to four, the observed reward is always in $[0, 4]$. The green dash line is the perfect prediction. This figure illustrate better generalization to unseen data for models which predict images. It also demonstrates the entanglement of exploration, task performance and prediction accuracy, when comparison with Figure 2 and Table 1.

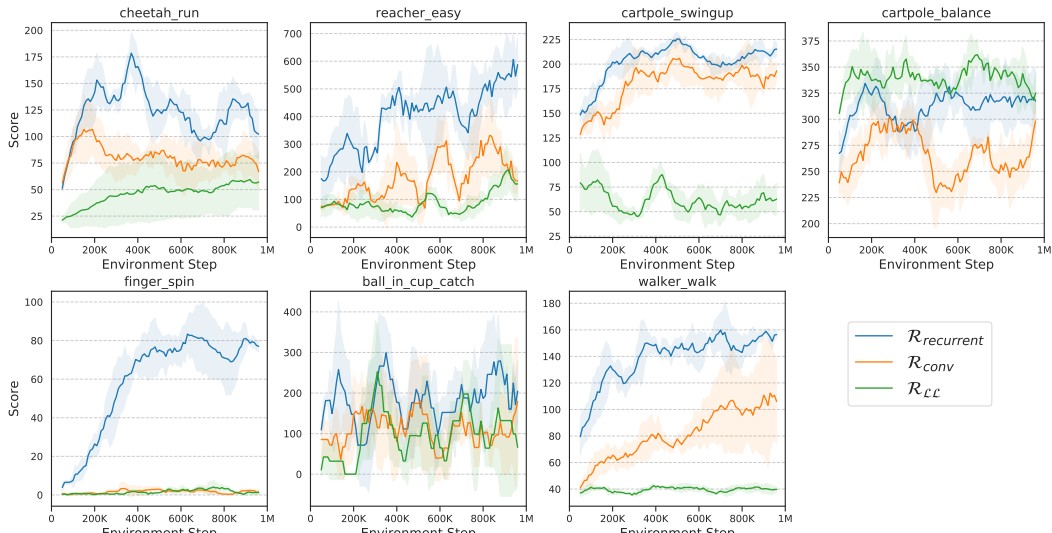

Figure 10: Comparison between different types of reward predictor models described in Section B. $\mathcal{R}_{\text{recurrent}}$: A simple recurrent convolutional network to predict the future rewards given the last observation (Table 8). This model has its own recurrent latent space which can be unrolled to predict future states. $\mathcal{R}_{\text{conv}}$: A simple convolutional network to predict the future rewards given the last observation (Table 9). This model predicts the rewards for the entire planning horizon in one shot. $\mathcal{R}_{\mathcal{LL}}$: The exact same model as $\mathcal{L}_{\mathcal{T}}\mathcal{L}_{\mathcal{R}}$ without image prediction. We finally used $\mathcal{R}_{\text{recurrent}}$ as $\mathcal{R}$ since it had the best overall performance, as can be seen in this figure.

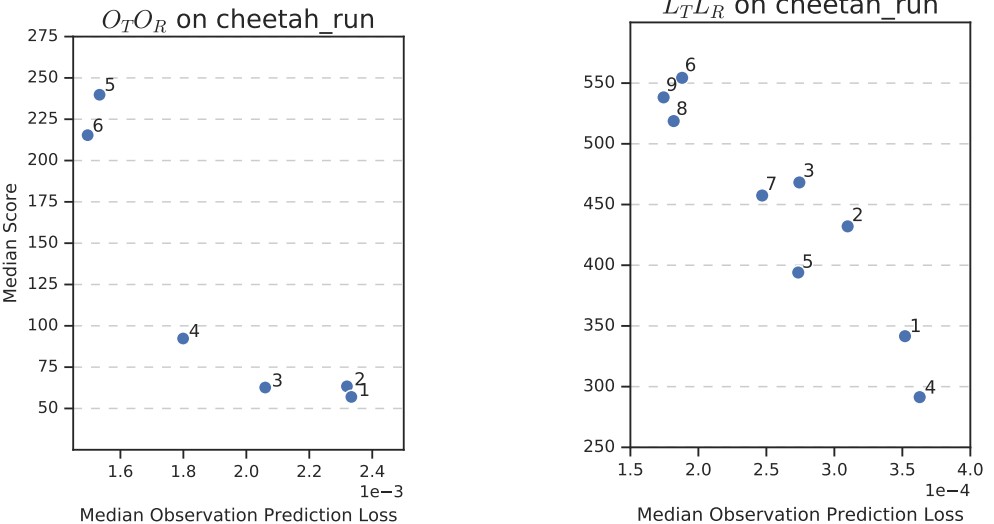

Figure 11: Correlation between accuracy of observation prediction and task performance, in the online setting. For each graph, we scaled down the indicated model (**left:** $\mathcal{O}_{\mathcal{T}}\mathcal{O}_{\mathcal{R}}$, **right:** $\mathcal{L}_{\mathcal{T}}\mathcal{L}_{\mathcal{R}}$) by multiple levels and report their median observation prediction error during training ($x$-axis) and median achieved score ($y$-axis). The annotation on each data point is the scale multiplier: higher means a bigger model (the detailed numbers can be found in appendix.) This graph clearly demonstrates the strong correlation between observation prediction accuracy and asymptotic performance.

## B  MODELS ARCHITECTURE AND DETAILS

To minimize the required hyper-parameter search, we reuse the existing stable implementations for the model designs in Figure 1. For other variations, we modified these existing models to match the new design. These models are inherently different in the tasks they do which means these models vary substantially their training and inference time. Table 7 summarized the characteristics of these models.

**$\mathcal{L}_\mathcal{T}\mathcal{L}_\mathcal{R}$: Latent transition - Latent reward**
This variant corresponds to several prior methods such as PlaNet (Hafner et al., 2018). Therefore, we directly use PlaNet as a representative model of this class (a description of this model can be found in Section B.2). By default, this model does back-propagate from both the reward loss and observation prediction loss into its learned latent space. Given the computational cost and for consistency, we had to use the same number of trajectories between models. This means we used a different set of planning hyper-parameters for PlaNet (particularly shorter training and prediction horizon of 12 vs 50 and lower number of trajectory proposals of 128 vs 1000). We had to make this change since other models are much slower than PlaNet to train and training these models with the original numbers from PlaNet would take over 2 months using eight v100s.

**$\mathcal{L}_\mathcal{T}\mathcal{O}_\mathcal{R}$: Latent transition - Observation reward**
Here, we again use the PlaNet architecture to model the dynamics in the latent space; however, we remove its reward predictor and instead utilize the reference reward model to predict the future reward given the predicted next observation. Following the design of PlaNet, we again back-propagate gradients from both the reward and observation.

**$\mathcal{O}_\mathcal{T}\mathcal{O}_\mathcal{R}$: Observation transition - Observation reward**
For this design, we used SV2P (Babaeizadeh et al., 2018) for modeling the dynamics in the observation space (details in Section B.1). Then, we use the reference reward model to predict the expected reward given the prediction observation. The main reason for using SV2P as the video model is its stability compared to other next frame predictors. For this model, we do not back-propagate gradients from the reward prediction into the latent space learned by observation prediction. We also did not use 3-stages of training and only performed the last stage of training (identical to PlaNet). According to the authors of SV2P, the three stage training is only required for highly stochastic and visually complex tasks. We verified the same performance with and without three stages of training on `cheetah_run`. To be clear, this means that all of our models follow the exact same training regime.

**$\mathcal{O}_\mathcal{T}\mathcal{L}_\mathcal{R}$: Observation transition - Latent reward**
Again, we use SV2P to model the dynamics in the observation space, however, the input to the reward predictor is the internal states of SV2P (i.e., the aggregated states of all Conv-LSTMs of the main tower described in (Babaeizadeh et al., 2018; Finn et al., 2016a)). This is the design behind some of the most recent works in visual model-based RL such as SimPLe Kaiser et al. (2020). However, we only back-propagate gradients from the observation prediction error into the learned representation.

**$\mathcal{R}$: Reward prediction without observation prediction**
We tried multiple reward predictors from images and chose the best one as $\mathcal{R}$:

1. $\mathcal{R}_{\text{recurrent}}$: A simple recurrent convolutional network to predict the future rewards given the last observation (Table 8). This model has its own recurrent latent space which can be unrolled to predict future states.

2. $\mathcal{R}_{\text{conv}}$: A simple convolutional network to predict the future rewards given the last observation (Table 9). This model predicts the rewards for the entire planning horizon in one shot.

3. $\mathcal{R}_{\mathcal{LL}}$: The exact same model as $\mathcal{L}_\mathcal{T}\mathcal{L}_\mathcal{R}$ without image prediction.

Figure 10 compares the performance of each one of these models on all the tasks. We finally used $\mathcal{R}_{\text{recurrent}}$ as $\mathcal{R}$ since it had the best overall performance.

**Oracle:** We use the environment itself as an oracle model. Since we are using the same planning algorithm across all of the models above, the rewards from the environment approximate the maximum possible performance with this planner.

| Model | Training Signal | Transition | Reward Input |
|-------|-----------------|------------|--------------|
| $\mathcal{R}$ | Reward Error | - | - |
| $\mathcal{O}_\mathcal{T}\mathcal{O}_\mathcal{R}$ | Reward Error + Observation Error | Observation Space | Predicted Observation |
| $\mathcal{O}_\mathcal{T}\mathcal{L}_\mathcal{R}$ | Reward Error + Observation Error | Observation Space | Latent States |
| $\mathcal{L}_\mathcal{T}\mathcal{L}_\mathcal{R}$ | Reward Error + Observation Error | Latent Space | Latent States |
| $\mathcal{L}_\mathcal{T}\mathcal{O}_\mathcal{R}$ | Reward Error + Observation Error | Latent Space | Predicted Observation |

Table 4: Summary of possible model designs based on whether or not to predict the future observations. All of the models predict the expected reward conditioned on previous observations, rewards and future actions. Moreover, the top four methods predict the future observations as well. These methods can model the transition function and reward function either in the latent space or in the observation space. Another design choice for these models is to whether or not share the learned latent space between reward and observation prediction.

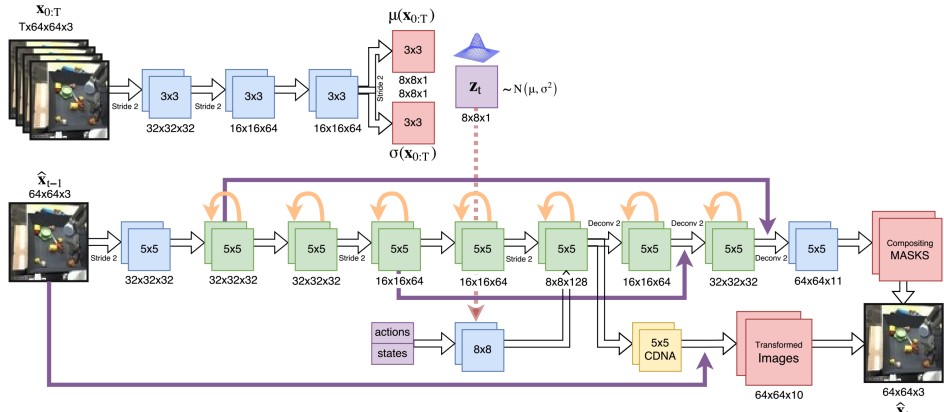

Figure 12: Architecture of SV2P. At training time, the inference network (top) estimates the posterior $q_\phi(\mathbf{z}|\mathbf{x}_{0:T}) = \mathcal{N}\big(\mu(\mathbf{x}_{0:T}), \sigma(\mathbf{x}_{0:T})\big)$. The latent value $\mathbf{z} \sim q_\phi(\mathbf{z}|\mathbf{x}_{0:T})$ is passed to the generative network along with the (optional) action. The generative network (from Finn et al. (2016a)) predicts the next frame given the previous frames, latent values, and actions. At test time, $\mathbf{z}$ is sampled from the assumed prior $\mathcal{N}(\mathbf{0}, \mathbf{I})$.

### B.1 SV2P ARCHITECTURE

Introduced in Babaeizadeh et al. (2018), SV2P is a conditional variational model for multi-frame future frame prediction. Its architecture, illustrated in Figure 12 consists of a ConvLSTM based predictor $p(\mathbf{x}_{c:T}|\mathbf{x}_{0:c-1}, \mathbf{z})$ where $\mathbf{z}$ is sampled from a convolutional posterior approximator assuming Guassian distribution $q_\phi(\mathbf{z}|\mathbf{x}_{0:T}) = \mathcal{N}\big(\mu(\mathbf{x}_{0:T}), \sigma(\mathbf{x}_{0:T})\big)$. The predictor tower is conditioned on the previous frame(s) and future proposed actions while the posterior is approximated given the future frame itself during the training. At inference, $\mathbf{z}$ is sampled from $\mathcal{N}(\mathbf{0}, \mathbf{I})$. This network is trained using the reparameterization trick (Kingma & Welling, 2014), and optimizing the variational lower bound, as in the variational autoencoder (VAE) (Kingma & Welling, 2014):

$$\mathcal{L}(\mathbf{x}) = -\mathbb{E}_{q_\phi(\mathbf{z}|\mathbf{x}_{0:T})}\big[\log p_\theta(\mathbf{x}_{t:T}|\mathbf{x}_{0:t-1}, \mathbf{z})\big] + D_{KL}\big(q_\phi(\mathbf{z}|\mathbf{x}_{0:T})||p(\mathbf{z})\big)$$

where $D_{KL}$ is the Kullback-Leibler divergence between the approximated posterior and assumed prior $p(\mathbf{z})$ which in this case is the standard Gaussian $\mathcal{N}(\mathbf{0}, \mathbf{I})$. Here, $\theta$ and $\phi$ are the parameters of the generative model and inference network, respectively. SV2P is trained using a 3-stage training regime which is only required for visually complex tasks and was not utilized in this paper. The list of hyper-parameter used can be found Table 11.

### B.2 PLANET ARCHITECTURE

Introduced by Hafner et al. (2018), PlaNet is a model-based agent that learns a latent dynamics model from image observations and chooses actions by fast planning in the latent space. To enable accurate

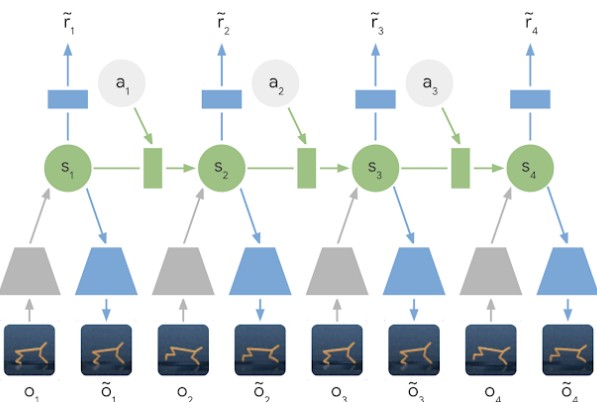

Figure 13: Architecture of PlaNet (Hafner et al., 2018). Learned Latent Dynamics Model: In a latent dynamics model, the information of the input images is integrated into the hidden states (green) using the encoder network (grey trapezoids). The hidden state is then projected forward in time to predict future images (blue trapezoids) and rewards (blue rectangle).

long-term predictions, this model uses stochastic transition functions while utilizing convolutional encoder and decoder as well as image predictors. The main advantage of PlaNet is its fast unrolling in the latent space since it only needs to predict future latent states and rewards, and not images, to evaluate an action sequence. PlaNet also optimizes a variational bound on the data log-likelihood:

$$\mathcal{L}(\mathbf{x}) = \sum_{t=1}^{T} \left( \underbrace{\underset{q(s_t|o_{\leq t}, a_{<t})}{\mathrm{E}}[\ln p(o_t|s_t)]}_{\text{reconstruction}} - \underbrace{\underset{q(s_{t-1}|o_{\leq t-1}, a_{<a-1})}{\mathrm{E}}\left[\mathrm{D}_{\mathrm{KL}}[q(s_t|o_{\leq t}, a_{<t}) \| p(s_t|s_{t-1}, a_{t-1})]\right]}_{\text{complexity}} \right).$$

We used the same training procedure and hyper-parameter optimization described in Hafner et al. (2018). The list of hyper-parameter used can be found Table 12.

It is worth mentioning that although we used PlaNet as $\mathcal{L}_{\mathcal{T}}\mathcal{L}_{\mathcal{R}}$ with no changes, our results are different from Hafner et al. (2018). This is because we used $128$ trajectories vs $1K$ in CEM (Table 10) as well as a training horizon of 12 vs 50 in Hafner et al. (2018) (Table 12). We made these changes to keep planning consistent across our models. Other models are slower than $\mathcal{L}_{\mathcal{T}}\mathcal{L}_{\mathcal{R}}$ to train and explore (Table 7) which made it infeasible to use them with original planning and training horizon of Hafner et al. (2018). Please check Section B.4 for our approximate compute cost.

### B.3 HYPER-PARAMETER SEARCH PROCEDURE

To get reasonable performance from each one of these models, we follow the same hyper-parameter search as Hafner et al. (2018); we grid search the key hyper-parameters on cheetah_run, selecting the best performing one, and then using the same set of parameters across all of the tasks. This means that these models will perform their best on cheetah_run. We are mainly doing this to keep the hyper-parameter search computationally feasible. The final used hyper-parameter can be found in Table 11 and Table 12. Contrary to Hafner et al. (2018), we did not optimize the action repeat per task and instead used the fixed value of four.

### B.4 APPROXIMATED COMPUTATION COST

In this paper we have three main experiments using eight models across seven tasks in two online and offline settings, where each run can take up to seven days on eight V100s (please check Table B.4 for a model break-down). That is 100K V100-hours (or $\sim \$70K$ at the time of writing of this paper) per random seed (we did three). Note that the cost is not only about training but also unrolling hundreds of random proposals per step. This is both time and computation expensive particularly for models with observation space transitions.

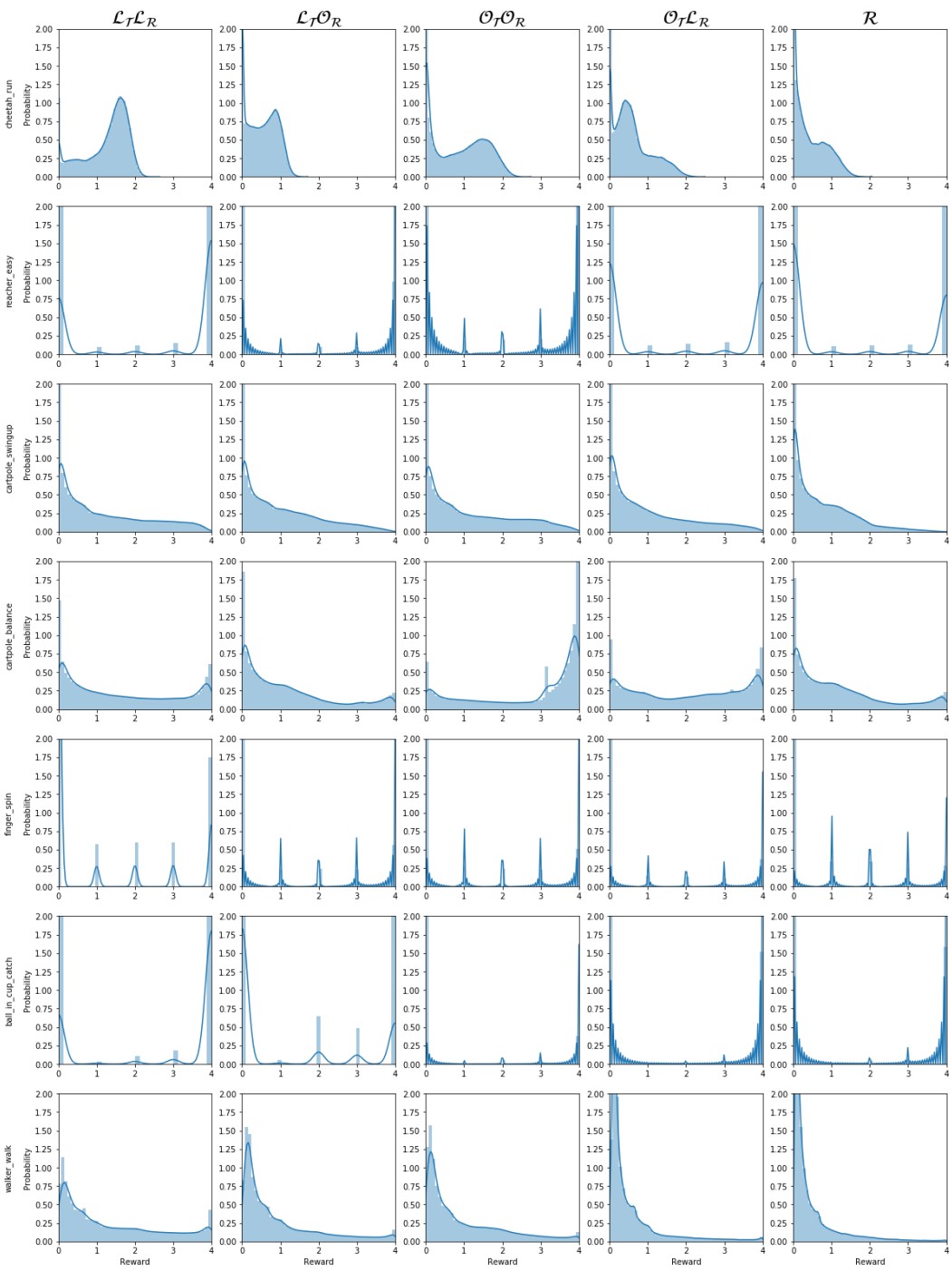

Figure 14: This figure illustrates the distribution of the rewards in trajectories collected by model variations in the online setting. Since the rewards for each frame is $[0, 1]$ and the action repeat is set to four, the observed reward is always in $[0, 4]$ ($x$-axis).

| | cheetah run | reacher easy | cartpole swingup | cartpole balance | finger spin | ball_in_cup catch | walker walk |
|---|---|---|---|---|---|---|---|
| **Online** | | | | | | | |
| $\mathcal{R}$ | $102 \pm 23$ | $588 \pm 95$ | $215 \pm 7$ | $317 \pm 31$ | $77 \pm 3$ | $205 \pm 75$ | $156 \pm 3$ |
| $\mathcal{O_T O_R}$ | $447 \pm 24$ | $480 \pm 106$ | $381 \pm 30$ | $849 \pm 111$ | $135 \pm 97$ | $43 \pm 37$ | $373 \pm 21$ |
| $\mathcal{O_T L_R}$ | $240 \pm 40$ | $801 \pm 60$ | $398 \pm 17$ | $575 \pm 10$ | $98 \pm 39$ | $204 \pm 71$ | $251 \pm 23$ |
| $\mathcal{L_T L_R}$ | $426 \pm 10$ | $768 \pm 135$ | $341 \pm 18$ | $445 \pm 16$ | $355 \pm 20$ | $711 \pm 177$ | $645 \pm 20$ |
| $\mathcal{L_T O_R}$ | $233 \pm 12$ | $757 \pm 91$ | $261 \pm 8$ | $343 \pm 41$ | $308 \pm 25$ | $421 \pm 144$ | $541 \pm 12$ |
| Oracle | $671$ | $85$ | $394$ | $595$ | $556$ | $957$ | $926$ |
| **Offline** | | | | | | | |
| $\mathcal{R}$ | $217 \pm 8$ | $996 \pm 7$ | $258 \pm 6$ | $415 \pm 7$ | $171 \pm 5$ | $985 \pm 20$ | $300 \pm 7$ |
| $\mathcal{O_T O_R}$ | $503 \pm 12$ | $981 \pm 16$ | $685 \pm 11$ | $978 \pm 11$ | $148 \pm 6$ | $910 \pm 19$ | $568 \pm 7$ |
| $\mathcal{O_T L_R}$ | $490 \pm 11$ | $999 \pm 9$ | $406 \pm 6$ | $923 \pm 12$ | $244 \pm 5$ | $949 \pm 18$ | $731 \pm 6$ |
| $\mathcal{L_T L_R}$ | $502 \pm 7$ | $994 \pm 10$ | $413 \pm 9$ | $543 \pm 8$ | $373 \pm 5$ | $996 \pm 19$ | $552 \pm 8$ |
| $\mathcal{L_T O_R}$ | $282 \pm 7$ | $993 \pm 15$ | $281 \pm 7$ | $411 \pm 7$ | $259 \pm 6$ | $932 \pm 19$ | $383 \pm 7$ |
| **Difference** | | | | | | | |
| $\mathcal{R}$ | $115$ | $408$ | $43$ | $98$ | $94$ | $780$ | $144$ |
| $\mathcal{O_T O_R}$ | $56$ | $501$ | $304$ | $129$ | $13$ | $867$ | $195$ |
| $\mathcal{O_T O_R}$ | $250$ | $198$ | $8$ | $348$ | $146$ | $745$ | $480$ |
| $\mathcal{O_T O_R}$ | $76$ | $226$ | $72$ | $98$ | $18$ | $285$ | $93$ |
| $\mathcal{O_T O_R}$ | $49$ | $236$ | $20$ | $68$ | $49$ | $511$ | $158$ |

Table 5: The asymptotic performance of various model designs in the online and offline settings and their differences. For the online setting the reported numbers are the average (and standard deviation) across three runs after the training. For the offline setting, the reported numbers are the same as Table 1 rounded up $\pm$ their standard deviation across three runs. This table demonstrates a significant performance improvement by predicting images almost across all the tasks. Moreover, there is a meaningful difference between the results for the online and the offline settings signifying the importance of exploration. Please note how some of best-performing models in the offline setting perform poorly in the online setting and vice-versa. This is clear from the bottom section of the table which includes the absolute difference of offline and online scores.

| | # Parameters | Training Step | Inference Step | #GPUs |
|---|---|---|---|---|
| $\mathcal{R}$ | 81.6K | 3 | 5 | $1 \times V100$ |
| $\mathcal{O_T O_R}$ | 8.34M | 40 | 310 | $8 \times V100$ |
| $\mathcal{O_T L_R}$ | 8.38M | 44 | 361 | $8 \times V100$ |
| $\mathcal{L_T L_R}$ | 5.51M | 32 | 10 | $1 \times V100$ |
| $\mathcal{L_T O_R}$ | 5.29M | 56 | 101 | $8 \times V100$ |

Table 6: Wall clock time of training and inference step of each model. The numbers are in seconds. Model $\mathcal{R}$ which only predicts the reward is the fastest model while the models with pixel space dynamics are the slowest. Please note that $\mathcal{R}$ is much smaller models compare to the other ones since it does not have any image decoder.

| | cheetah run | reacher easy | cartpole swingup | cartpole balance | finger spin | ball_in_cup catch | walker walk |
|---|---|---|---|---|---|---|---|
| $\mathcal{R}$ | 20.4 | 117.6 | 43.0 | 63.4 | 15.4 | 41.0 | 31.2 |
| $\mathcal{O_T O_R}$ | 1.4 | 1.5 | 1.2 | 2.7 | 0.4 | 0.1 | 1.2 |
| $\mathcal{O_T L_R}$ | 0.7 | 2.2 | 1.1 | 1.6 | 0.3 | 0.6 | 0.7 |
| $\mathcal{L_T L_R}$ | 42.6 | 76.8 | 34.1 | 44.5 | 35.5 | 71.1 | 64.5 |
| $\mathcal{L_T O_R}$ | 2.3 | 7.5 | 2.6 | 3.4 | 3.0 | 4.2 | 5.4 |

Table 7: Cost-normalized scores foe each model. These numbers are the online score achieved by each model divided by the inference cost (time) of the corresponding model. As expected, faster models – i.e. $\mathcal{R}$ and $\mathcal{L_T L_R}$ which do not predict the images at inference time – get a big advantage. This table clearly shows that $\mathcal{L_T L_R}$ is generally a good design choice if there is no specific reason for modeling the dynamic or reward function in the pixel space.

| Layer Type | Filters | Size | Strides | Activation |
|---|---|---|---|---|
| Last Observation Encoder | | | | |
| Convolutional | 32 | 3x3 | 2 | LeakyReLU |
| Convolutional | 64 | 3x3 | 2 | LeakyReLU |
| Convolutional | 16 | 3x3 | 2 | LeakyReLU |
| Convolutional | 8 | 3x3 | 2 | LeakyReLU |
| Action Encoder | | | | |
| Dense | 32 | - | - | LeakyReLU |
| Dense | 16 | - | - | LeakyReLU |
| Dense | 8 | - | - | LeakyReLU |
| Last Reward Encoder | | | | |
| Dense | 32 | - | - | LeakyReLU |
| Dense | 16 | - | - | LeakyReLU |
| Dense | 8 | - | - | LeakyReLU |
| Next State Predictor | | | | |
| LSTM | 64 | - | - | LeakyReLU |
| Reward Predictor | | | | |
| Dense | 32 | - | - | relu |
| Dense | 2 | - | - | relu |
| Dense | 1 | - | - | None |

Table 8: The architecture of $\mathcal{R}_{\text{recurrent}}$.

| Layer Type | Filters | Size | Strides | Activation |
|---|---|---|---|---|
| Last Observation Encoder | | | | |
| Convolutional | 32 | 3x3 | 2 | LeakyReLU |
| Convolutional | 64 | 3x3 | 2 | LeakyReLU |
| Convolutional | 16 | 3x3 | 2 | LeakyReLU |
| Convolutional | 8 | 3x3 | 2 | LeakyReLU |
| Action Encoder | | | | |
| Dense | 32 | - | - | LeakyReLU |
| Dense | 16 | - | - | LeakyReLU |
| Dense | 8 | - | - | LeakyReLU |
| Last Reward Encoder | | | | |
| Dense | 32 | - | - | LeakyReLU |
| Dense | 16 | - | - | LeakyReLU |
| Dense | 8 | - | - | LeakyReLU |
| Reward Predictor | | | | |
| Dense | 32 | - | - | LeakyReLU |
| Dense | 16 | - | - | LeakyReLU |
| Dense | planning horizon | - | - | None |

Table 9: The architecture of $\mathcal{R}_{\text{conv}}$

| Parameter | Value |
|---|---|
| Planning Horizon | 12 |
| Optimization Iterations | 10 |
| Number of Candidate Trajectories | 128 |
| Number of Selected Trajectories | 12 |

Table 10: The hyper-parameters used for CEM. We used the same planning algorithm across all models and tasks.

| Hyper-parameter | Value |
|---|---|
| Input frame size | 64×64×3 |
| Number of input frames | 2 |
| Number of predicted frames | 12 |
| Beta $\beta$ | 1e-3 |
| Latent channels | 1 |
| Latent minimum log variance | -5.0 |
| Number of masks | 10 |
| ReLU shift | 1e-12 |
| DNA kernel size | 5 |
| Optimizer | Adam |
| Learning Rate | 1e-3 |
| Epsilon | 1e-3 |
| Batch Size | 64 |
| Training Steps per Iteration | 100 |
| Reward Loss Multiplier | 1.0 |

Table 11: The hyper-parameter values used for SV2P (Babaeizadeh et al., 2018; Finn et al., 2016a) model (used in $\mathcal{O}_\mathcal{T}\mathcal{O}_\mathcal{R}$ and $\mathcal{O}_\mathcal{T}\mathcal{L}_\mathcal{R}$). The number of ConvLSTM filters can be found in Table 14. The rest of parameters are the same as Babaeizadeh et al. (2018).

| Hyper-parameter | Value |
|---|---|
| Input frame size | 64×64×3 |
| Number of predicted frames | 12 |
| Units in Deterministic Path | 200 |
| Units in Stochastic Path | 30 |
| Units in Fully Connected Layers | 300 |
| Number of Encoder Dense Layers | 1 |
| Number of Reward Predictor Layers | 3 |
| Free Entropy nats | 3.0 |
| Minimum Standard Deviation | 0.1 |
| Optimizer | Adam |
| Learning Rate | 1e-3 |
| Epsilon | 1e-3 |
| Batch Size | 50 |
| Training Steps per Iteration | 100 |
| Reward Loss Multiplier | 10.0 |

Table 12: The hyper-parameter values used for PlaNet (Hafner et al., 2018) model (used in $\mathcal{L}_\mathcal{T}\mathcal{O}_\mathcal{R}$ and $\mathcal{L}_\mathcal{T}\mathcal{L}_\mathcal{R}$). The rest of parameters are the same as Hafner et al. (2018).

| Hyper-parameter | Value |
|---|---|
| Input frame size | 64×64×3 |
| Number of predicted rewards | 12 |
| Number of Reward Predictor Layers | 3 |
| Number of Reward Predictor Layers | 3 |
| Minimum Standard Deviation | 0.1 |
| Optimizer | Adam |
| Learning Rate | 1e-3 |
| Epsilon | 1e-3 |
| Batch Size | 32 |
| Training Steps per Iteration | 100 |

Table 13: The hyper-parameter values used for $\mathcal{R}_{\text{recurrent}}$ (used as $\mathcal{R}$). The hyper-parameters for the planner is the same as other models (Table 10).

| Model ID | ConvLSTM 1 | ConvLSTM 2 | ConvLSTM 3 | ConvLSTM 4 | ConvLSTM 5 | ConvLSTM 6 | ConvLSTM 7 |
|---|---|---|---|---|---|---|---|
| 1 | 1 | 1 | 2 | 2 | 4 | 2 | 1 |
| 2 | 2 | 2 | 4 | 4 | 8 | 4 | 2 |
| 3 | 4 | 4 | 8 | 8 | 16 | 8 | 4 |
| 4 | 8 | 8 | 16 | 16 | 32 | 16 | 8 |
| 5 | 16 | 16 | 32 | 32 | 64 | 32 | 16 |
| 6 | 32 | 32 | 64 | 64 | 128 | 64 | 32 |

Table 14: The downsized version of $\mathcal{O}_\mathcal{T}\mathcal{O}_\mathcal{R}$. We down-scaled the model by reducing the number of ConvLSTM filters, limiting the modeling capacity and thereby potentially increasing their prediction error. The detailed architecture of the model and layers can be found in Finn et al. (2016a); Babaeizadeh et al. (2018).

| Model ID | Units in Fully Connected Layers | Units in Deterministic Path |
|---|---|---|
| 1 | 100 | 200 |
| 2 | 100 | 100 |
| 3 | 100 | 50 |
| 4 | 50 | 200 |
| 5 | 50 | 100 |
| 6 | 50 | 50 |
| 7 | 200 | 200 |
| 8 | 200 | 100 |
| 9 | 200 | 50 |

Table 15: The downsized version of $\mathcal{L}_\mathcal{T}\mathcal{L}_\mathcal{R}$. We down-scaled the model by reducing the number of units in fully connected paths, limiting the modeling capacity and thereby potentially increasing their prediction error. The detailed architecture of the model and layers can be found in Hafner et al. (2018).

