# OpenReview forum: "On Trade-offs of Image Prediction in Visual Model-Based Reinforcement Learning"
_ICLR.cc/2021/Conference — Reject_

### Official Review · AnonReviewer2 · 2020-10-27
**Questionable empirical setup and problems with the statistical significance of the results**

**Rating:** 4
**Confidence:** 4

**Review:**

Summary

This paper empirically investigates the effect of different reward-prediction models for model-based reinforcement learning (MBRL) in the particular context of decision-time planning with the cross-entropy method and visual tasks. To that end, five models are evaluated: four models that predict rewards and observations jointly, while one model only learns to predict rewards but without observation prediction. The authors claim that predicting observations in addition to rewards improves performance of MBRL in their setting (in terms of cumulative reward maximization). This hypothesis is tested on a bunch of tasks from the DeepMind Control Suite, both in an online and offline setting. The authors conclude, based on their experiments, that their hypothesis can be confirmed.


Quality and Details

My main problem is that I don't entirely understand the particular constellation of models to be compared with each other to begin with---see Figure 1. To my understanding, Figure 1 just depicts two different classes of models: the four on the left and the one on the right. Note however that this becomes clear only after studying the appendix (Appendix B) in more detail, since the models are not described properly in the main text. The four models on the left can be subdivided into 2 further subclasses depending on whether transitions are modelled in latent or observed space: latent-variable models refer to the PlaNet architecture and observation-space models to the SV2P architecture (each can be trained with and without additional reward prediction). The stand-alone model on the right is a specific model not related to PlaNet or SV2P.

I believe the question of whether additional observation prediction helps can be investigated for each of the 2 top-level classes separately. For example, the left four models could be amended to only predict rewards---running PlaNet or SV2P but stripping away the observation prediction component? Looking at appendix Figure 13 that describes PlaNet, the observation component could be ignored in the course of training such as to only predict rewards. Similarly in Appendix Figure 12 that describes SV2P, the autoregressiveness could happen at the hidden layer without observation input/output. In a same way could the stand-alone model for reward prediction be amended to enable observation prediction.

After studying the appendix in more detail, I came to realize that the authors actually followed one of the ideas above and trained PlaNet with only reward prediction but not observation prediction. They found that this performs worse compared to another reward-only-prediction baseline (which they finally report in the main paper). Concluding, the presentation is quite convoluted and it is still missing whether SV2P can be amended to predict rewards only, or whether the best-performing reward-prediction baseline can be extended to predict observations.

A clear execution and presentation of the setting described above would allow to draw conclusions about whether predicting observations actually helps *across* different architectures, but not the setting studied by the authors that uses separate architectures for joint reward-observation prediction and reward-only prediction.

Additionally, assuming the experimental setup makes sense (which I feel it does not entirely), I am still concerned about the statistical significance of the experimental results from Section 4.2 on online learning. There are just 3 seeds per experiment---so it is not clear which joint reward-observation model is best, although it seems indicative that the reward-only prediction model is worse across tasks compared to the joint models. Statistical significance is only getting worse in Section 4.3 which studies the offline setting: because here only one performance value is reported *at the end of training* as opposed to *in the course of training* (making it even more difficult to judge the statistical significance, since RL algorithms are known to produce rapidly performance-changing policies in the course of training).


Clarity

The clarity of the paper is low. While multiple different prediction models are studied, none of them are presented in detail in the main paper, neither regarding architecture nor learning objective. The appendix provides more information, revealing how complex the models are. These details do matter in an empirical study and a missing description in the main paper makes it even harder to judge the conclusions drawn from the experiments.


Originality and Significance

Investigating the necessity of observation prediction (in addition to reward prediction) on RL performance is interesting. However, the paper does not entirely convince me regarding both the experimental setup and statistical significance.


Pros

A lot of environments are tested.


Cons

The experimental setup is questionable and so is the statistical significance of the results.


Minor

I believe the caption of Figure 3 draws conclusions about which regions of the state space have been visited. However, only empirical reward distributions are presented, i.e. I don't see how one can draw conclusions about state-space visitation from there.

Also, the main online results from Figure 2 show in some environments that the oracle baseline (that knows the true environment dynamics) is worse compared to the MBRL algorithms. The authors mention in the text that this is due to a limited planning horizon---I still do not understand why the MBRL algorithms work better given that they also need to cope with a limited planning horizon.

---

> ### Author Response · Authors · 2020-11-13
> **Response to AnonReviewer2 (part2)**
>
> ***
> ***I still do not understand why the MBRL algorithms work better given that they also need to cope with a limited planning horizon***
>
> Thank you for asking this important question. The easiest setting that this phenomena can be explained in is an environment with sparse rewards out of the planning horizon (e.g. reacher_easy in our experiments). The planning method (CEM) chooses the trajectory with highest reward from all the randomly proposed trajectories, but the Oracle (correctly) predicts only zeros for all of them. This means that the final selected trajectory will be random until the goal reaches within the planning horizon in which the Oracle will predict a non-zero reward. However, other models are not enforced to predict zeros and ones and they can guide the planning method by predicting higher reward for the trajectories that get the agent *closer* to the goal. In other words, this is mainly because of smoother non-binary prediction of the rewards by other models and it will be different if someone enforces a hard prediction.
>
> ***
> ***I am still concerned about the statistical significance of the experimental results from Section 4.2 on online learning***
>
> Regarding three random seeds, we whole-heartedly agree with this problem. Unfortunately, we are limited by the number of experiments that we can run and adding more random seeds would be challenging given the computational cost. Please refer to Section B.4. for an approximation of this cost.
>
> Regarding offline results, similar to the online setting, the results are actually the average of three different runs (the achieved score at the end of the training). We updated the caption of Table 1 to clarify this. We also added standard deviations of these results (Table 5 in the appendix) to help with the statistical significance. However, we also would like to emphasize that our goal was not to compare these designs to the last digit of achieved score, but rather, we are interested in the ball-park of the results that they can get to and how this behaviour changes from offline to online and vice versa.
>
> ***
> ***While multiple different prediction models are studied, none of them are presented in detail in the main paper***
>
> Thank you for raising this point. As you mentioned, there are many details and we agree that these details matter a lot, for evaluation of the work as well as expanding it. And we would like to thank you for taking the time to look into the appendix. As we mentioned in the introduction, given the empirical nature of this work, our observations are only as good as our implementations and therefore details matter. Unfortunately, we were limited by the number of pages in this manuscript and we chose to use this space for analysis of the results which left us no space for the implementation details and had to move them into the appendix. Please let us know if you think this order should be reversed.

---

> > ### Comment · AnonReviewer2 · 2020-11-19
> > **Re: Response to AnonReviewer2**
> >
> > Thanks to the authors for their detailed response. Unfortunately, after reading all the reviews (and the responses), i became confident in my assessment that this is not a paper suited for the ICLR main track. I agree with Reviewer #4 and have decreased my score to 4 accordingly. I do not agree at all with Reviewers #1 and #3 in their positive assessment of the manuscript.
> >
> > The main concern that remains, as also outlined by others, is that the model line up chosen for comparison is not fair. This is in line with Reviewer #4 who asks for a "backbone model" to start comparisons from, and also has concerns regarding the reward-only-prediction baseline. In that line, Reviewer #1 also questions the reward-prediction baseline as a "straw man model". I do understand how expensive it is to run experiments in order to obtain statistically significant results, and how frustrating it can be reviewers asking for that without proper reason. In this manuscript however, since new theoretical or conceptual ideas are absent, the quality needs to be judged based on the experimental side which is not convincing in order to arrive at the authors' claim that image prediction helps reinforcement learning. These aspects combined with the lacking clarity and presentation of the paper makes a reject inevitable (in my opinion).

---

> > > ### Author Response · Authors · 2020-11-20
> > > **Re: Re: Response to AnonReviewer2**
> > >
> > > Thank you for the feedback.
> > >
> > > First, we would like to emphasize that **the main claim of the paper (importance of image prediction) is tested in both settings, with and without a backbone model**. Figure 2 and Table 1 compare the performance of different models, while Figure 4 and Figure 6 provide a backbone model comparison: it is PlaNet as a backbone model with and without image decoder. These figures also clearly demonstrate that our base $R$ performs better that a modified backbone, as we mention in our response to AnonReviewer1.
> > >
> > > Regarding using a single back-bone model: while we could have implemented such a backbone model, it seems likely that our work could then be criticized for the same issues such as fairness in comparison. For example, we could start with PlaNet as the base model and then add/remove/modify components to create variations of the model to compare it to. e.g. we could remove the image decoder to change the model to $R$. Or we could modify the latent transition to make it compatible with pixel space and move the reward function to top of the decoder to create a $O_T O_R$. Now the question is whether this approach will address the raised concerns or not? We will argue otherwise for mainly two reasons:
> > > 1. The variations of the backbone model would be still substantially different, particularly in terms of number of parameters. e.g. adding a big decoder and a big transition function will make the model much much bigger with more parameters. Or removing the image decoder will make the model much smaller in terms of parameters. This means that this variation will be substantially different which will cause the same concerns regarding fairness.
> > > 2. It will make the models way less compatible and comparable with previous work. Currently, a main concern in the reviews is that our only proposed model $R$ is the "straw man model", despite a full chapter in the appendix on tested architectures. Another concern by AnonReviewer4 is that our models are not aligned with SOTA. Switching to a back-bone model and its variations would make these concerns much more severe.
> > >
> > > Therefore, both options have shortcomings and we decided in the balance that our approach was the lesser of these two issues. We did our best to make these comparisons fair and comparable to SOTA by using pre-existing models and working closely with the corresponding authors for any modification we had to make which makes us confident in our model selection.  We also tried to be crystal clear about this, particularly in the last paragraph of 4.1 which we clearly discuss that "there are going to be numerous shortcomings for ANY implementation" and we still believe that there is no perfect solution for such study.
> > >
> > > Finally, regarding the statistical significance, we agree that more random seeds are better, however, given the current number of trials we can still calculate (from the numbers in Table 5 or using the actual performance values) the p-value for each experiment using a two-tailed t-test. For example the hypothesis that $L_T L_R$ has higher performance as $R$ on Cheetah-Run has a t-value of ~39 and the p-value is < .00001 which makes it strongly significant. In fact, **from 28 possible comparisons to $R$ (4 models x 7 tasks) 20 have a p-value < 0.05**. We will include these numbers in the final version of the paper.
> > >
> > > We hope the detailed response above illustrates the soundness of our experimental setup. Please let us know whether this helped resolve your concern.

---

> ### Author Response · Authors · 2020-11-13
> **Response to AnonReviewer2 (part1)**
>
> We would like to thank the reviewer for their in-depth review and their great questions. We provided detailed answers to all of your questions below. In particular, we added a new Table 5 which includes the standard deviation of our results to address your concern regarding statistical significance. Additionally, we updated the text of the paper to increase clarity. We hope that these answers and updates fully address your concerns but please let us know if there are other issues that we should address.
>
> ***
> ***My main problem is that I don't entirely understand the particular constellation of models to be compared with each other to begin with---see Figure 1.***
>
> Thank you for your detailed review. An easier way of looking at this is that we are investigating different possible designs given three axes of variation:
> 1. Whether or not the model predicts images. This is the first partition that you mentioned.
> 2. Dynamic model space (latent space or pixel space). This is the second partition that you mentioned which in our implementations corresponds to PlaNet (latent space dynamics) and SV2P (pixel space dynamics).
> 3. Reward prediction input space (latent space or pixel space). This will be $L_T L_R$ and $O_T L_R$ in contrast with $L_T O_R$ and $O_T O_R$.
>
> Given these three axes of variations there should be eight (2x2x2) possible designs however 2 and 3 only make sense when images are being predicted. e.g. the input to the reward prediction model cannot be the predicted image because there is no predicted image. That leaves us with the five designs in Figure 1.
>
> There is another way of looking at this as well as mentioned by AnonReviewer1. The critical difference is which loss is being back-propagated and how. This directly corresponds to which targets are being predicted (i.e. next image and/or reward) and how the gradients from these signals are being back-propagated. We updated the caption of Figure 1 to clear this up.
>
> ***
> ***I believe the question of whether additional observation prediction helps can be investigated for each of the 2 top-level classes separately. For example, the left four models could be amended to only predict rewards---running PlaNet or SV2P but stripping away the observation prediction component?***
>
> Thank you for raising this important point. As you pointed out, Figure 4 and 6 do exactly your suggestion for PlaNet. Unfortunately, this experiment cannot be repeated for any setting other than $L_T L_R$ in a meaningful way. This is because other designs model either the dynamics or the reward function in the pixel space and given that there is a **direct dependency on the predicted image** removing the image prediction doesn’t make sense — such a model could not predict forward or could not predict rewards. For example SV2P ($O_T$) models the dynamics in the pixel space: not predicting the future image means there will be no future image to be used for predicting the one after that.
>
> Another way of thinking about it is that removing the image prediction part of the models with $O_T$ or $O_R$ will change their category into another design (because it will move the design from $O_X$ to $L_X$) which leaves us with $L_T L_R$ as the only one that this experiment can be done for.
>
> ***
> ***only empirical reward distributions are presented, i.e. I don't see how one can draw conclusions about state-space visitation from there.***
>
> Thank you for the great question. We agree that the relation between the actual state-space visitation and the distribution of the observed rewards is a not one-to-one function. This means that two agents can explore different regions of the state-space while having the same reward distribution, but, if they do have a different reward distribution it means they explored different regions of the state space, at least in frequency. That is the only point that Figure 3 is trying to make, that $L_T L_R$ and $O_T O_R$ managed to visit higher reward space more frequently compared to $R$.

---

### Official Review · AnonReviewer4 · 2020-10-28

**Rating:** 3
**Confidence:** 3

**Review:**

In this paper, the authors compare several kinds of MBRL models, including observation prediction, to show which approach is better. In conclusion, The models that contain prediction on both a reward and an observation outperform the ones with only reward prediction.

[Quality]

This paper requires a revision for enhancing clarity. For example, Section 4 seems exceedingly long, and it contains too many redundant details, which should not have been emphasized.

[Originality & Significance]

This paper provides neither a new method nor a different perspective.

[Strengths]
+ The authors showed the effect of using visual observation prediction for MBRL models. It might show the way for further MBRL models.
+ It was mentioned on the paper that prediction accuracy of observations and rewards and exploration performance is on the trade-off relationship.

[Weaknesses]
- The number of parameters on the model that predicts only rewards is much (about a hundred times) smaller than other models. Additionally, the authors used SV2P and PlaNet to implement each experiment; it is doubtable that comparisons are fair enough.
- There are no clear and consistent differences between models with an observation prediction, which reduces this work's novelty. It is better to use four different SOTA models or four modified models from one backbone model instead of modifying two models.
- The trade-off relationship is mentioned, but there is not enough quantitative and theoretical analysis to show this point.

[Comment]
- There is no bolded item in Table 2; it seems to be omitted, or the result is not strong enough.

---

> ### Author Response · Authors · 2020-11-13
> **Response to AnonReviewer4**
>
> We would like to thank the reviewer for their review. We updated the text (including the caption of Figure 1 which introduces the designs as well as our discussion in the Conclusions) to increase clarity and added multiple new tables to the paper to summarize our results. Please see our detailed response below and let us know if this addresses your concerns, or if there are other issues that we should address. We are glad to provide clarifications for any further questions you may have.
>
> ***
> ***This paper provides neither a new method nor a different perspective.***
>
> We respectfully disagree. Although our paper does not introduce any new method, it presents new perspectives and insights. As mentioned by AnonReviewer1, our paper provides “additional insight about exploration vs modeling efficiency” as well as “additional experiments that separates exploration efficiency”. We believe the insights in this paper can help future researchers by providing a new perspective on how to think about model based RL from images. For a complete list of our proposed claims please refer to the list at the end of the introduction.
>
> ***
> ***This paper requires a revision for enhancing clarity. For example, Section 4 seems exceedingly long, and it contains too many redundant details, which should not have been emphasized.***
>
> Thank you for your review. We updated the text (including the caption of Figure 1 which introduces the designs as well as our discussion in the Conclusions) to increase clarity. Our paper is an experimental paper, and Section 4 includes all the experiments and their results. We tried to improve its readability by breaking it down into multiple sub-sections and building each experiment on top of the previous one. Please let us know if this addresses your concerns.
>
> ***
> ***The number of parameters on the model that predicts only rewards is much (about a hundred times) smaller than other models. Additionally, the authors used SV2P and PlaNet to implement each experiment; it is doubtable that comparisons are fair enough.***
>
> Thank you for raising this point. We addressed this issue in the “Implementation and Hyper-Parameter tuning” paragraph of Section 4.1. These models are inherently different from each other in what they model and how they model it. Clearly a model that predicts 64x64x3 images has to have more parameters than a model that predicts a single reward number. Please refer to the same paragraph for a discussion for why we believe our work adequately addresses this potential issue.
>
> ***
> ***There are no clear and consistent differences between models with an observation prediction, which reduces this work's novelty. It is better to use four different SOTA models or four modified models from one backbone model instead of modifying two models.***
>
> To the best of our knowledge, our models are aligned with SOTA, as also mentioned by AnonReviewer1. As far as we know, some of these designs such as $L_T O_R$ are not explored before but these are possible designs when one considers all possible cases given the mentioned axes of variation in the model design. We would happily switch to SOTA implementation of these designs for DeepMind control tasks if the reviewer points us to them.
>
> Regarding consistency, that is one of the reasons for reusing the same implementation whenever possible which is also pointed out by AnonReviewer1 as a strength of the paper. Please note that modifying one back-bone model will still result into different models with substantially different number of parameters and behavior given the fact that their design would be different. We are happy to hear the reviewer's thoughts on how we can improve this consistency.
>
> ***
> ***The trade-off relationship is mentioned, but there is not enough quantitative and theoretical analysis to show this point.***
>
> We are more than happy to address these issues if the reviewer points them out more clearly. Other reviewers mention that the claims in this paper are mostly well-supported, which we agree with, but we are always interested in improving this support.
>
> ***
> ***There is no bolded item in Table 2; it seems to be omitted, or the result is not strong enough.***
>
> We purposefully omitted bolding the lower numbers in Table 2 to discourage the readers from only looking at them, and instead encourage the readers to consider the correlation. In other words, this table does not include a winner-takes-it-all comparison and the goal of this paper is to highlight that there is no clear winner (as mentioned by AnonReviewer1), hence no bolding. There is still a strong correlation between these numbers and Table 1 which is one of the claims of the paper and the strength of this claim can be judged independently.

---

### Official Review · AnonReviewer1 · 2020-10-28
**Useful taxonomy and benchmarks; generality of results w.r.t. taxonomy overstated**

**Rating:** 6
**Confidence:** 2

**Review:**

The paper proposes a simple taxonomy of objectives for training predictive models for planning in visual model-based RL, and evaluates them in a set of consistent experimental tasks and using a largely consistent set of models. It claims that in the context of the models investigated, backpropagating losses based on future observations yields better performance than using only the reward as training signal, and that prediction accuracy of future observations is also more predictive of task performance than reward prediction errors. It additionally empirically demonstrates the magnitude of effect exploration policies can have in this setting, showing how a lot of the models that perform worse online simply explore less well.

I think the paper is largely easy to read and understand, and systematic/structured exploration of modeling choices is important especially given the steady trickle of results showing that specific modeling choices can be less important than how well-tuned the implementations are. The additional insight about exploration vs modeling efficiency is well-supported and the additional experiment that separates exploration efficiency is well-thought out. I also think it's great that less-conclusive or less-succesfful experiments (e.g. on optimism) are still included in the appendix rather than file-drawered, though I'm puzzled that the claim regarding implicit optimism is made at the end of section 4.3. considering what appendix A2 shows.

That said, I have some additional comments about ways in which the paper could be improved:

1. I think the paper overstates the generality of its results, considering the usage of only one specific choice of each architecture and loss (with the exception of the R-model) and one planner. The choices seem to be reasonable and consistent with the SOTA, but they're also only one way of setting up each kind of loss in the taxonomy, and the paper doesn't make a strong argument for why they should be general. Especially given how the paper shows that there's no clear uniform winner across tasks, one could reasonably expect that the variability across other variants not explored is similarly large. Similarly, the underperforming oracle highlights the contribution of the fixed-horizon planner to the overall results, and calls into question the claim in section 4.1 about how the oracle approximates optimal performance. I think it's good that the architectures were kept consistent across variants when possible, but convergent evidence (even from scaled-down toy examples) would help make the point, in combination with more caveats about the generality.

2. A related concern is whether the reward prediction models are genuine competitors or straw man models, since the remaining architectures / models come from previously published results and I'm not sure if the reward prediction models do (I think $\mathcal{R}_{LL}$ is similar to what was used by Havens et al. 2019, though that was a non-archival NeurIPS workshop paper). It's surprising that the paper does not take any of the prior work it cites as the exemplar model for reward prediction, similarly to how it does with pixel prediction. On the other hand, if these models are credible competitors from past work, that work should be cited.

3. The focus in the writing on what the model predicts is a bit misleading in that unless I'm missing something, the critical difference is which loss is being backpropagated. In order to plan, I need to simulate rollouts, so even some of the R-models have a next state predictor (except if they predict the full horizon at once, as in $\mathcal{R}_{conv}$), and the reward-loss models can have worse reward predictions, presumably by having less training signal (e.g. Tab 2). This could be clarified in the text and Figure 1.

4. The overall results are ultimately "messy" in the sense that there's no clear winners or patterns. This in itself is worth knowing (and a point that the paper mostly embraces). That said, there are a few missed opportunities for further insights, including:
    - Performance across different kinds of difficulties. The paper makes the point early on that its test environments focus on different difficulties (contact discontinuities, large state spaces, long-term memory, long-term planning, sparse rewards). This point isn't revisited -- some discussion along these dimensions might be useful, beyond "different training signals win on different tasks, except reward prediction, which never does".
    - The paper makes the point that using reward only as training signal can be substantially cheaper (section 4.4), and that latent-dynamics models are faster than pixel-space dynamics models (table 5). Does this mean we might expect cost-normalized performance to show some interesting patterns?

5. It would be useful if a visual comparison between the online results (Fig 2) and offline results (Tab 1) could be made, either by including Tab 1 results in panels on the figure, or moving the table to the appendix in favor of a figure that compares offline and online results. Similarly, the tables do not facilitate the comparison made in the caption to Tab. 2 regarding the relationship between reward prediction accuracy and task performance.

[I think the paper provides a useful case study and I appreciate the caveats added to the conclusion, but based on additional discussion with the other reviewers, I think that the paper's more general claims remain unsupported and insufficiently moderated. I am reducing my rating accordingly.]

---

> ### Author Response · Authors · 2020-11-13
> **Response to AnonReviewer1**
>
> We would like to thank the reviewer for their in-depth review as well as their great suggestions to improve the paper. As suggested, we included a new Table 5 in the appendix which includes all the numbers, offline and online, for completeness and better comparison. Also, newly added Table 7, includes a cost-normalized score as you proposed. Additionally, we updated the text of the paper (including the caption of Figure 1) to increase clarity. Please let us know if this fully addresses your concerns, or if there are other issues that we should address.
>
> ***
> ***Does this mean we might expect cost-normalized performance to show some interesting patterns?***
>
> We included a cost (compute time) normalized score to address this comment. Please take a look at Table 7 in the appendix. As expected, faster models -- i.e. $R$ and $L_T L_R$ which do not predict the images at inference time -- get a big advantage.
>
> ***
> ***It would be useful if a visual comparison between the online results (Fig 2) and offline results (Tab 1) could be made.***
>
> Thank you for raising this point. AnonReviewer3 also raised the same point. To address this issue, we added Table 5 in the appendix which includes detailed results of online and offline scores as well as their differences. This table clearly shows the performance difference of some of the models with and without exploration.
>
> ***
> ***I'm puzzled that the claim regarding implicit optimism is made at the end of section 4.3. considering what appendix A2 shows.***
>
> Thank you for raising this point. Our statements at the end of Section 4.3 are mostly speculative. We updated the text to reflect this more clearly to match it with the results in the appendix.
>
>
> ***
> ***It's surprising that the paper does not take any of the prior work it cites as the exemplar model for reward prediction, similarly to how it does with pixel prediction.***
>
> Unfortunately, we could not find any reward only model that fits into our experimental setup and works on DeepMind Control tasks from pixels. Our cited references rely on either extra tricks (e.g. Sekar et al 2020 which uses novelty based exploration) or extra predictions (e.g. in VPN networks of  Oh et al 2017), That being said, our reward model is based on the reward model of PlaNet (Hafner et al 2018) which was designed and optimized for DeepMind Control tasks. We spent a lot of time optimizing it to make it work even better without image prediction. This improvement is visible if you compare the red curve from Figure 4 (the one with no image prediction) and $R$ from Figure 2.
>
> Can you please provide a link to the paper you mentioned (Havens et al. 2019)? We can try to provide a comparison.
>
> ***
> ***I think the paper overstates the generality of its results, considering the usage of only one specific choice of each architecture and loss (with the exception of the R-model) and one planner.***
>
> We 100% agree with the reviewer and acknowledge that this is one of the main limitations of our study. As you mentioned, the lower performance of the Oracle model is for sure caused by the limitation of the planning method and our observations could have been different if we used a policy optimization method instead. We also acknowledge that there are many other axes of variation that can be explored. This paper constitutes the first step toward such analysis and in the writing of the paper we tried really hard to make sure that we are not over generalizing to the cases that are not covered by our experiments.
>
> We updated the last paragraph of Conclusions to emphasize on this point and make sure we do not claim strong generalization to other cases.
>
> ***
> ***The focus in the writing on what the model predicts is a bit misleading in that unless I'm missing something, the critical difference is which loss is being backpropagated***
>
> This is correct. We do indeed equate the choice of loss (which is back-propagated into the model) with the choice of prediction target -- i.e., all training signals come from the prediction target.
>
> We updated the caption of Figure 1 to highlight this point.
>
> ***
> ***Performance across different kinds of difficulties. ***
>
> We would like to sincerely thank the reviewer for raising this point. Our goal was to test our hypotheses on a diverse set of environments but we hesitated to discuss different kinds of difficulties mainly because of the generalization issue that you mentioned. We can speculate that the agents which model the dynamics of the environment in the pixel space perform poorly on more visually complex tasks in which details matter such as ball_in_cup_catch and finger_spin (and it is supported by our results), however, we are concerned with over-stating our findings without providing additional support for such claims. Instead, we decided to discuss these details whenever we could provide enough data e.g. why predicting images is important.

---

> > ### Comment · AnonReviewer1 · 2020-11-19
> > **Thank you for addressing my comments.**
> >
> > The Havens et al. 2019 paper is https://arxiv.org/abs/1912.04201, Learning Latent State Spaces for Planning through Reward Prediction.
> >
> > W.r.t. the cost-normalized score, it's not surprising that the cost of predicting pixels blows out any other differences, but it's a bit more interesting that LtOr outperforms the other pixel models on all tasks in cost-normalized score. Related to this, is the caption meant to say that LtLr is generally a good design choice, or LtOr? Among the reward prediction models it seems like the normalized performance is a toss-up depending on task.

---

> > > ### Author Response · Authors · 2020-11-25
> > > **Re: Thank you for addressing my comments.**
> > >
> > > ***is the caption meant to say that LtLr is generally a good design choice, or LtOr?***
> > >
> > > Thank you for pointing this out. That is correct. The caption should say $L_T L_R$ is the generally good design choice.
> > >
> > > ***
> > > ***The Havens et al. 2019 paper***
> > >
> > > Thank you for referring us to this paper. It seems it focuses on low-dim inputs which is different from our setup (pixels as the input space). There is, however, one image-based experiment in the Appendix Section C where the authors used frame stacking to get full observability (to quote: "two images per state to insure that velocity can be inferred and fully-observable assumption"). Unfortunately, this is also different from our setup which assumes partial observability that makes it hard to compare with our models. Regardless, we will make sure to cite it in the related works.
> > >
> > > ***
> > > Please let us know if this fully addresses your followup questions.

---

### Official Review · AnonReviewer3 · 2020-10-28
**Official Blind Review #3**

**Rating:** 7
**Confidence:** 4

**Review:**

- Summary:
    - This paper presents a study on the trade-offs of image prediction for model based RL
    - They find that image prediction loss is important and, surprisingly, reward prediction accuracy can be negatively correlated performance while image prediction accuracy is considerably more well correlated.
- Stengths
    - Clear presentation of results and overall well written
    - Comprehensive series of results that all build on one another
    - A large amount of care given to the implementations. The sentence "Given the empirical nature of this work, our observations are only as good as our implementations." is very apt.
    - The authors do a good job at pointing out the caveats and shortcomings of their study.
- Weaknessses
    - While a fair number of environments are studied, they are all visually very similar (all 3rd person camera, same background, etc.).  Adding in a very different environment, i.e. Visdoom, would strength the analysis.  While I don't think adding a different environment is necessary, this should be pointed out in the paper.
    - While the presentation is overall quite clear, I do have some suggestions (see bellow)
    - Questions:
        - How was training stopped in the off-line setting?
        - Were the same hyper-parameters used for both online and offline?
- Suggestions for improvement
    - A version of Table 1 with the online results at convergence would be great as the delta between online and offline performance is interesting.
    - A Pearson correlation coefficient and/or a Wilcoxon signed ranked test on the ranking induced by Table 1 vs. Table 2 for each environment would be a nice summary statistic.  Although the dynamic range of the values may not be large enough for this to be sensible.
    - If possible, color code the variant names in the tables with their line colors in plots.  Also, order the table the same as the figure legends.
- Overall
    - This paper presents an interesting, well-motivated, and comprehensive study on the trade-offs in image prediction for model based RL that also points to future directions of research.


## Post Rebuttal

I thank their authors for their response.  I have decided to maintain my rating, overall I still think this is a good paper that presents an interesting set of results.  The result that image prediction accuracy correlated better with asymptotic performance than reward prediction accuracy continues to intrigue me.  My fellow reviewers have some concerns that while I don't agree, I think they could be avoided with some changes in the presentation to the paper:

* The models used.  I understand the decision to move the model description to the appendix (not everything is going to fit in 8 pages), but the main paper would benefit from paragraph or two describing why which models were chosen and then referring to the proper places in the appendix for the full details.

* Reward-only model.  There was considerable concern about how much smaller that models is than the others.  Additional results showing that one with a comparable number of parameters does as poorly if not worse would be beneficial.  The 0% line in Fig 6 and Fig 8 is very similar to this hypothetical (if it exactly) so I believe this will pan out as expected.

* Presentation of results.  In my read, the most interesting result, anti-correlation between reward prediction accuracy and asymptotic performance, comes at the end.  Without that result, there is a way to read the paper as "yes image prediction helps, it increases the amount of supervision given to the model".  That result and the difference between online and offline shows that there is something unexpected going on here so perhaps leading with those and/or highlighting them more would help.

---

> ### Author Response · Authors · 2020-11-13
> **Response to AnonReviewer3**
>
> Thank you for the great review as well as constructive suggestions, comments and questions.
> As suggested, we reordered the table rows and color coded them to match the graphs. We also added the Pearson correlation coefficient to Table 2 for a better comparison with Table 1. We also included a new Table 5 in the appendix which includes all the numbers, offline and online for completeness. Additionally, we updated the text of the paper (including caption of Figure 1) to increase clarity, detailed below. Please let us know if this fully addresses your concerns, or if there are other issues that we should address.
>
> ***
> ***How was training stopped in the off-line setting?***
>
> Number of training iterations required for each episode is one of the hyper-parameters that we tuned for different models in the online setting. For the offline setting, we used the same number multiplied by the number of training episodes. This number is the “Training Steps per Iteration” in the hyper-parameter tables (Table 11 and Table 12 in the appendix). We also added Table 13 which included the hyper-parameters for model R. Please note that the training batch size also varies from model to model which is included in the same tables.
>
> ***
> ***Were the same hyper-parameters used for both online and offline?***
>
> Yes. We only optimized the hyper-parameters for online run of Cheetah-Run and then used the same set of parameters for the offline setting as well as other online tasks. Please look at Section B.3 (in appendix) for more details.
>
> ***
> ***While a fair number of environments are studied, they are all visually very similar (all 3rd person camera, same background, etc.). Adding in a very different environment, i.e. Visdoom, would strength the analysis.***
>
> We agree that trying the same experiments on visually different environments and particularly visually more complex tasks in an interesting direction to expand this study. Unfortunately, adapting all of the methods to an entirely new environment, including tuning of hyper-parameters and running multiple seeds for all methods, would be quite difficult, and surely does not fit into ICLR rebuttal time frame. Some of our results ($O_T O_R$ specifically) take multiple days to finish. But this is a direction that we are considering, as mentioned in the last paragraph of the conclusions.
>
> ***
> ***A version of Table 1 with the online results at convergence would be great as the delta between online and offline performance is interesting.***
>
> Thank you for your suggestion. The same suggestion is also raised by AnonReviewer1. We added Table 5 (in the appendix) which summarizes the online and offline results as well as their differences for better comparison.
>
> ***
> ***A Pearson correlation coefficient and/or a Wilcoxon signed ranked test on the ranking induced by Table 1 vs. Table 2 for each environment would be a nice summary statistic. Although the dynamic range of the values may not be large enough for this to be sensible.***
>
> Thank you for the great suggestion. We updated Table 2 to include the Pearson correlation coefficient which demonstrates a strong correlation between reward prediction error and task performance in the offline setting. Please note that In cases which all models are close to the maximum possible score of 1000 (such as ball_in_cup_catch) the correlation can be misleading because a better prediction does not help the model anymore.
>
> ***
> ***If possible, color code the variant names in the tables with their line colors in plots. Also, order the table the same as the figure legends.***
>
> Thank you for the suggestion. We reordered and recolored the tables to match the graphs.

---

### Decision · Program_Chairs · 2021-01-07
**Final Decision**

**Decision:**

Reject

**Comment:**

The authors compare different model based R algorithms to see whether observation prediction is important. They show that, as expected, it is. On the other hand, they seem to show that latent space prediction is not very useful. The study is limited to domains with image data: Does this domain have something particularly special? Perhaps experiments with smaller-scale POMDP problems might actually have shown something different. It is very difficult to do a study of this type properly, and although the authors have tried, it's hard to see how this paper can be accepted. I agree with some the positive points some reviewers have raised, but I think that, at the end of the day, the paper is trying to draw too general conclusions from a handful of datapoints. Were I writing this paper, I would first try the simplest version of the hypothesis with very basic environments that are, however, more varied than the ones shown here. Would the hypothesis hold, I'd scale up to more complex environments and try to also run with more seeds to get a clearer signal.